# Air–Sea Fluxes of Greenhouse Gases and Oxygen in the Northern Benguela Current Region During Upwelling Events

Eric J. Morgan[1,2], Jost V. Lavric[1], Damian L. Arévalo-Martínez[3], Hermann W. Bange[3], Tobias Steinhoff[3], Thomas Seifert[1], and Martin Heimann[1,4]

[1]Max Planck Institute for Biogeochemistry, Jena, Germany
[2]Now at: Scripps Institution of Oceanography, La Jolla, CA, U.S.A.
[3]GEOMAR Helmholtz Centre for Ocean Research, Kiel, Germany
[4]Institute for Atmospheric and Earth System Research (INAR) / Physics, University of Helsinki, Finland

**Correspondence:** Eric J. Morgan (ejmorgan@ucsd.edu)

**Abstract.** Ground-based atmospheric observations of $CO_2$, $\delta(O_2/N_2)$, $N_2O$, and $CH_4$ were used to make estimates of the air–sea fluxes of these species from the Lüderitz and Walvis Bay upwelling cells in the northern Benguela region, during upwelling events. Average flux densities ($\pm 1\sigma$) were $0.65 \pm 0.4$ $\mu$mol m$^{-2}$ s$^{-1}$ for $CO_2$ , $-5.1 \pm 2.5$ $\mu$mol m$^{-2}$ s$^{-1}$ for $O_2$ (as APO), $0.61 \pm 0.5$ nmol m$^{-2}$ s$^{-1}$ for $N_2O$, and $4.8 \pm 6.3$ nmol m$^{-2}$ s$^{-1}$ for $CH_4$. A comparison of our top-down (*i.e.*, inferred from atmospheric anomalies) flux estimates with shipboard-based measurements showed that the two approaches agreed within $\pm 55\%$ on average. During the study, upwelling events were sources of $CO_2$, $N_2O$, and $CH_4$ to the atmosphere. $N_2O$ fluxes were fairly low, in accordance with previous work suggesting that the evasion of this gas from the Benguela is smaller than for other Eastern Boundary Upwelling Systems (EBUS). Conversely, $CH_4$ release was quite high for the marine environment, a result that supports studies that indicated a large sedimentary source of $CH_4$ in the Walvis Bay area. These results demonstrate the suitability of atmospheric time series for characterizing the temporal variability of upwelling events and their influence on the overall marine GHG emissions from the northern Benguela region.

## 1 Introduction

Coastal margins, particularly those associated with the upwelling of nutrient-rich subsurface waters, are biogeochemically active regions (Levin et al., 2015). The air–sea fluxes of greenhouse gases (GHGs; referring to the long-lived greenhouse gases $CO_2$, $N_2O$, and $CH_4$) from or to such systems can vary markedly, both spatially and temporally (Torres et al., 1999; Naqvi et al., 2010; Evans et al., 2011; Reimer et al., 2013; Capone and Hutchins, 2013). This is because the occurrence and intensity of coastal upwelling events are episodic in nature, as they are forced by surface winds that occur under specific synoptic conditions; even large events happen only on a time scale of days (Blanke et al., 2005; Goubanova et al., 2013; Desbiolles et al., 2014a, b). The sporadic nature of upwelling implies that observations made during short-term campaigns may not capture the full range of flux variability.

Upwelled water is usually colder than the surrounding surface water, which means that the solubility of dissolved gases will decrease with increasing temperature as water masses warm at the surface. A competing influence for $CO_2$ exists in

that the supply of inorganic nutrients from an upwelling event can lead to blooms of phytoplankton and a net drawdown of atmospheric $CO_2$. For $O_2$, the ventilation of deeper water masses can drive a net flux into the ocean, or net productivity can create oversaturation of dissolved $O_2$. Hence, coastal upwelling regions can oscillate between being sources and sinks of $O_2$ and $CO_2$ (Torres et al., 1999; Santana-Casiano et al., 2009; González-Dávila et al., 2009; Gregor and Monteiro, 2013; Cao et al., 2014; Evans et al., 2015). Most coastal upwelling systems are also known to be regional hotspots of $N_2O$ emissions (Bange et al., 2001; Lueker et al., 2003; Cornejo et al., 2006; Bianchi et al., 2012; Arévalo-Martínez et al., 2015; Babbin et al., 2015). Air-sea fluxes of $CH_4$ are less constrained, but may be a significant term in the marine $CH_4$ budget (Rehder et al., 2002; Sansone et al., 2001).

A well-established method of estimating budgets of air–sea fluxes for GHGs in upwelling regions is to take wind fields and interpolated or representative surface measurements, use them to calculate a flux density, and scale it up over a selected area. The high variability of air–sea exchange means that determining budgets of air–sea fluxes of GHGs is challenging without a high degree of spatial and temporal sampling. We refer to this in the text as the "bottom-up" approach.

Another approach, and one that sidesteps some of these difficulties, is to use a "top-down method", *i.e.*, using atmospheric measurements to infer fluxes from the surface, using simple models (Lueker et al., 2003; Lueker, 2004; Nevison et al., 2004; Thompson et al., 2007; Yamagishi et al., 2008) or more complex inverse methods (*e.g.*, Rödenbeck et al. (2008)). A simple top-down approach has been successfully employed to detect air–sea fluxes of $CO_2$, $O_2$, and $N_2O$ from the California Current region from a coastal atmospheric monitoring station at Trinidad Head, California (Lueker et al., 2003; Lueker, 2004; Nevison et al., 2004). This work motivated our own efforts to see if anomalies related to upwelling events could be seen in continuous observations from an atmospheric measurement site located near the upwelling region in the northern Benguela, which is one of the least sampled EBUS for air–sea fluxes of GHGs (Nevison et al., 2004; Naqvi et al., 2010; Laruelle et al., 2014).

In this study, two years of continuous observations from a ground-based atmospheric observatory for greenhouse gases, the Namib Desert Atmospheric Observatory (NDAO), were utilized to create top-down estimates of the air–sea flux densities of $CO_2$, $O_2$, $N_2O$, and $CH_4$ from the Lüderitz and Walvis Bay upwelling cells, during upwelling events. We focus on individual upwelling events as we expect them to be distinguishable from other sources of intraseasonal variability based their apparent stoichiometry in the atmosphere, and because there are relatively few observation-based studies from this region, relative to other EBUS. This area of the coastal shelf, stretching from ca. 22°S to 28°S, is subject to the strongest surface winds and upwelling fluxes of water in the region, and surface chlorophyll is at a minimum (Lutjeharms and Meeuwis, 1987; Hagen et al., 2001; Demarcq et al., 2007; Veitch et al., 2009; Hutchings et al., 2009). These estimates were then compared with shipboard measurements from a cruise in the two upwelling centers.

## 2 Methods

### 2.1 Atmospheric Measurements at the Namib Desert Atmospheric Observatory

Continuous measurements of $CO_2$, atmospheric oxygen, $N_2O$, $CH_4$, and CO were made at the Namib Desert Atmospheric Observatory (NDAO), a background site located at 23.563°S, 15.045°E, in the central Namib Desert, at Gobabeb Research

and Training Centre. A full description of the measurement system is given in Morgan et al. (2015). In brief, observations were made at 21 m height above ground level with a Picarro ESP-1000 cavity ringdown spectrometer (CRDS) for $CO_2$ and $CH_4$, a Los Gatos $N_2O$/CO-23d cavity-enhanced absorption spectrometer for $N_2O$ and CO, and an Oxzilla FC-II dual absolute and differential oxygen analyzer for $\delta(O_2/N_2)$.

While the Picarro and Los Gatos analyzers measure continuously (data are recorded approximately 0.5 and 1 Hz, respectively), the Oxzilla analyzer measures the difference between sample air and a reference tank, at a data rate of 0.01 Hz. Calibration of the instruments was done through four working secondary standards and instrument performance was periodically checked with "target" cylinders (i.e., tanks of known mole fraction which are regularly remeasured). Reference gases were comprised of dry ambient air and stored in 50 L aluminum cylinders. Calibrations were run every 123 hours for the Picarro and
Los Gatos analyzers, and every 71 hours for the Oxzilla. All measurements were tied to primary standards on the following scales: WMO X2007 for $CO_2$, NOAA 2004 for $CH_4$, NOAA 2006a for $N_2O$, WMO X2004 for CO, and the Scripps Institute for Oceanography S2 scale for $\delta(O_2/N_2)$.

The average uncertainty for each species during the study period was 0.028 ppm for $CO_2$, 6.5 per meg for $\delta(O_2/N_2)$, 0.21 ppb for $N_2O$, 0.17 ppb for $CH_4$, and 0.15 ppb for CO. $CO_2$, $N_2O$, $CH_4$, and CO are all expressed as dry air mole fractions, in
ppb or ppm (1 ppm = 1 $\mu$mol mol$^{-1}$; 1 ppb = 1 nmol mol$^{-1}$).

By convention, changes atmospheric oxygen were quantified as the $O_2/N_2$ ratio relative to a standard, in per meg units (Keeling and Shertz, 1992):

$$\delta(O_2/N_2) = \left( \frac{(O_2/N_2)_{sample}}{(O_2/N_2)_{ref}} - 1 \right) \times 10^6 \tag{1}$$

In order to isolate the influence of air-sea exchanges on $O_2/N_2$, we have employed the use of a data-derived tracer, known as
atmospheric potential oxygen (APO), which masks variations of $O_2/N_2$ that are due to terrestrial biosphere exchange (Stephens et al., 1998). Variations in APO are thus primarily due to fossil fuel burning and air-sea gas exchange of $O_2$. APO is defined as:

$$APO = \delta(O_2/N_2) + \frac{1.1}{X_{O_2}}(CO_2 - 350) \tag{2}$$

Here 1.1 is the nominal global average oxidative ratio (OR; $\Delta O_2/\Delta CO_2$ on a mol/mol basis) for terrestrial photosynthesis and
respiration (Severinghaus, 1995), $X_{O_2}$ is the mole fraction of oxygen in the atmosphere, 0.2095, as defined by the Scripps $O_2$ Program scale. 350 is a reference $CO_2$ value, $CO_2$ is the in situ concentration of carbon dioxide, in ppm. APO and $\delta(O_2/N_2)$ are both expressed in per meg. In some instances it is useful to express APO in molar units that are of the same relative magnitude as the dry air mole fraction in ppm. This conversion is done by multiplying $\delta(O_2/N_2)$ or APO by $X_{O_2}$. We refer to this unit as parts per million equivalents, or "ppm eq".

## 2.2 Remote Sensing Data

Following the reasoning of Goubanova et al. (2013), sea surface temperature (SST) data were obtained from the Remote Sensing Systems (http://www.remss.com/) data archive. The Tropical Rainfall Measuring Mission (TRMM) Microwave Imager

(TMI) daily optimally interpolated SST product was selected. The major advantage of this instrument is its ability to measure SST through clouds, which are considerable over the coast, as the TMI measures frequencies in the microwave region (4–11 GHz). The drawback of this dataset is that there is no data within 100 km of the coast. Large upwelling features, however, extend much farther out than this and are readily seen by TMI (Goubanova et al., 2013). Even with this loss of near-shore data, the data coverage is still superior to that of optical sensors like the Moderate-Resolution Imaging Spectroradiometer (MODIS). The TRMM SST data presented in this work is deseasonalized by subtracting a second harmonic fit to the data, as it showed a strong seasonal cycle which masked some of the intraseasonal variability when plotted as a time series.

Wind speed data for the South Atlantic was also derived from the TMI instrument on the TRMM satellite. This dataset is a level 3 product which gives the 10 m wind speed over marine areas within the sensor's field of view. The 18.7 GHz channel data product was selected. Like the SST data, a major drawback of this dataset is the absence of data within 100 km of land.

Two surface chlorophyll products were used, both level 3 binned products that combined data from multiple satellites, accessed through ESA GlobColour website (http://www.globcolour.info/). The first is denoted CHL1-GSM; this dataset is a merged product of two different sensors (during the time period considered), MODIS and the Visible Infrared Imaging Radiometer Suite (VIIRS). The data is merged using the Garver, Siegel, Maritorena (GSM) model, which blends the normalized water-leaving radiances instead of the end product (chl-*a* concentrations) (Maritorena and Siegel, 2005). The second product is denoted CHL1-AVW; these data are merged using a weighted average method (AVW). Like CHL1-GSM, it combines data from MODIS and VIIRS for the time frame considered.

## 2.3 Atmospheric Back-Trajectories

Back-trajectories, which trace the path of a particle from a receptor point backwards in time, provide a history of recent atmospheric transport. Back-trajectories were simulated with the the HYbrid Single Particle Lagrangian Integrated Trajectory (HYSPLIT) 4 model (Draxler and Hess, 1997, 1998) from NDAO for 120 hours, for the entire time series. A new trajectory was calculated every six hours, starting at 0:00 UTC, i.e., at 1:00, 7:00, 13:00, and 19:00 local time. The model was run with a spatial resolution of $1° \times 1°$ and a temporal resolution of 1 hour. A vertical cut-off of 10 km a.g.l. was used. Meteorological fields were obtained from the National Center for Environmental Prediction (NCEP) Global Data Assimilation System (GDAS).

## 2.4 Identification of Upwelling Events and Selection of Atmospheric Anomalies

A subset of the coastal region was selected to represent the Lüderitz and Walvis Bay upwelling cells. The boundaries of this domain were at 13°E, 15°E, 23°S, and 27°S (Figure 1, *right panel*), representing an ocean area of 56,196 km$^2$. This domain covers the continental shelf and a small portion of the continental slope; the mean water depth is 390 m. We selected this domain because it represented an area of the coast where strong upwelling occurs regularly (Demarcq et al., 2007), where this upwelling was spatially distinct from other upwelling cells reported in the literature (Lutjeharms and Meeuwis, 1987; Veitch et al., 2009), and where upwelling was downwind of the station during upwelling events. These criteria were considered desirable because they would provide the best opportunities for relating atmospheric anomalies to upwelling events. These determinations were based on analysis of our SST dataset, and HYSPLIT back-trajectories.

Upwelling events were identified based on SST anomalies and 10 m wind speed anomalies. An event was determined to occur if the average deseasonalized SST of the domain was 0.5°C or lower than a smoothed, second-degree polynomial fit to the entire time series, and the average 10 m wind speed of the study area was 2.5 m s$^{-1}$ above a smoothed, second-degree polynomial fit to the wind data. These thresholds were determined through visual inspection of maps and time series of SST and wind speed data, and are specific to the domain chosen, since the data considered were averages of the entire area. One standard deviation of the SST anomaly was 0.92°C, and one standard deviation of the wind speed anomaly was 3.1 m s$^{-1}$. Since the resolution of the SST and wind speed time series was daily, all higher resolution data falling within a day during which an upwelling event occurred was similarly flagged.

Atmospheric anomalies associated with upwelling events were quantified against a baseline determined through a second-harmonic fit that was generated iteratively to all data, excluding all points that lay outside 1 standard deviation from the curve for the subsequent iteration. Small adjustments were made to the anomaly to raise or lower the curve in cases where it did not intersect with background values within ± 5 days of the event. The NDAO data was filtered by wind (wind speeds greater than 2 m s$^{-1}$ and wind direction within the NNW–SSW sector), as well as by back-trajectory to exclude anomalies which were not associated with marine air masses. For the latter, trajectories could not reside for more than 36 hours of the total 120 hours over land, and could not travel more than 50 km inland past NDAO. A final data selection step was to exclude days that had CO values greater than 15 ppb above baseline, to remove any time periods that may have been influenced by biomass burning. The filtered time series, with terrestrial influences removed, is shown in Figure 2.

## 2.5  Top-Down Air–Sea Flux Estimates

In order to estimate the surface flux associated with atmospheric anomalies due to upwelling events, the approach of Lueker et al. (2003) was adopted. A simple model was employed to describe the change in the concentration of a species within a well-mixed column of air as it moves over a source region (Jacob, 1999):

$$\Delta C = \begin{cases} \frac{F}{hq}\left(1 - e^{-\frac{qx}{U}}\right), & \text{for } 0 \leq x \leq L \\[2mm] \Delta C_L \left(e^{-\frac{q(x-L)}{U}}\right), & \text{for } x \geq L \end{cases} \tag{3}$$

Here $\Delta C$ is the concentration of the species of interest, in mol m$^{-3}$, expressed as an anomaly against the background. $\Delta C$ is a function of $x$, which is the distance to the observation point from the flux region. $L$ is the point at which the column (with height $h$, in m) leaves this region, characterized by a constant flux, $F$, in mol m$^{-2}$ hr$^{-1}$, and a constant wind speed, $U$, in m hr$^{-1}$. After the column leaves the flux region ($x \geq L$, *i.e.* when the coast is reached), the loss of $\Delta C$ from its peak at $L$ ($\Delta C_L$) is governed by dilution due to mixing of background air. This requires the dilution rate constant, $q$, in hr$^{-1}$, to be known.

We solved the equation for $F$ by taking the other variables as follows: $\Delta C$ was determined from our atmospheric record (see Section 2.4), wind speeds ($U$) were obtained from satellite data (Section 2.2), and $h$ was taken as the average height of the planetary boundary layer (PBL) for the Lüderitz/Walvis Bay domain over the course of any given event. PBL data was acquired

from the European Centre for Medium-Range Weather Forecasting's (ECMWF) ERA-Interim dataset (Dee et al., 1979). $x$ was taken as the length traveled along a back-trajectory from NDAO to the area affected by upwelling.

The dilution rate constant, $q$, was estimated by comparing measurements of $CO_2$ and $CH_4$ made during the RV *Meteor* cruise M99 (see Section 2.7). Back-trajectories from NDAO were matched to the closest ship location at the appropriate time. Any back-trajectory points that were within 100 km of the ship—both horizontally and vertically—within the space of 1 hour were identified, and a dilution rate constant was calculated for both $CO_2$ and $CH_4$, as (Price et al., 2004):

$$q = \frac{1}{t} \ln \left( \frac{C_{M99} - C_b}{C_{NDAO} - C_b} \right) \tag{4}$$

Where $C_{M99}$ and $C_{NDAO}$ are the mole fractions of $CO_2$ or $CH_4$, as measured in situ on the *Meteor* or at NDAO, and $C_b$ is taken from the fit to the baseline of the NDAO time series for either species, as described in Section 2.4. $t$ is the travel time along the back-trajectory, in hr. While this assumes that there is no flux the gas between the vessel and the station, we excluded instances when the anomaly was heavily influenced by surface fluxes by filtering for poor agreement between $CO_2$ and $CH_4$. Fluxes of these gases are not well correlated, and can go in opposite directions for summertime air-sea fluxes, so we assume that it is unlikely that both values of $q$ would be biased by the same amount. From 32 values of $q$ we excluded all but 13 for poor agreement (excluding determinations with $|q_{CO_2} - q_{CH_4}| > 0.01$). The average ($\pm 1\sigma$) was taken to arrive at a single value, $0.011 \pm 0.006 \, \text{hr}^{-1}$. Since this is still a rough approximation, we also present estimates of the estimated flux, $F$, with a value of $q$ tuned to provide the best agreement with an in situ flux estimate (see Section 2.8).

## 2.6 Estimated Top-Down Flux Density Uncertainty

Uncertainties were propagated in quadrature, *e.g.*:

$$\sigma_x = \sqrt{\sigma_a^2 + \sigma_b^2} \tag{5}$$

for operations involving addition or subtraction ($x = a + b$), and

$$\sigma_x = \sqrt{\left( \frac{\sigma_a}{a} \right)^2 + \left( \frac{\sigma_b}{b} \right)^2} \cdot x \tag{6}$$

for operations involving multiplication or division ($x = ab$), where $\sigma$ is the uncertainty for a given variable. The uncertainty in $\Delta C$ for each species was the propagated analytical uncertainty (see Section 2.1) of the difference between the average of $C$ during the maximum of the anomaly and the baseline, plus a small uncertainty term for the value of the baseline itself based on the standard deviation of 1,000 simulations in which one third of data were discarded and the curve fit repeated. $x$ and $L$ were both taken as the standard deviation of the term calculated independently for eight HYSPLIT back-trajectories from NDAO within 0-50 hours back from the anomaly detection. The uncertainty for $U$ was one standard deviation of wind speeds for the study domain within the 50 hr period.

## 2.7 In Situ Measurements During RV *Meteor* Cruise M99

Cruise M99 of the RV *Meteor* left Walvis Bay on July 31$^{st}$, 2013, and returned to port on August 23$^{rd}$. Continuous or semi-countinuous measurements of atmospheric $CO_2$, $N_2O$, and $CH_4$, as well as dissolved $CO_2$, $O_2$, and $N_2O$, were conducted throughout the cruise. Flask samples were taken for discrete measurements of $\delta(O_2/N_2)$.

Atmospheric measurements of $CO_2$ and $CH_4$ were made with a CRDS analyzer (model G1301, Picarro Inc, Santa Clara, CA, USA) located in the atmospheric chemistry laboratory. The instrument's internal pump was used to draw air through a 7 m length of 1/4" SERTOflex tubing, at a flow rate 150 mL min$^{-1}$. Inlets identical to those used at NDAO (Morgan et al., 2015) were placed on the starboard railing of the 6$^{th}$ superstructure deck, just above the atmospheric chemistry lab, at a total height of $\sim$21 m above sea level. A second-order, instrument-specific water correction was performed in lieu of physical or

chemical drying, identical to the method described in Morgan et al. (2015). As the instrument's pressure control seemed to be affected by strong vessel motion, measurements were excluded if the cavity pressure deviated by more than 0.04 torr from the setpoint of 140 torr. Calibrations were conducted on average every three days and target measurements were made once per day. Reference gases were calibrated at MPI-BGC GASLAB. The uncertainty, derived from the target measurements as at NDAO, was determined to be $\pm$ 0.03 ppm for $CO_2$ and $\pm$ 0.43 ppb for $CH_4$. The dataset was filtered for contamination by the

ship's exhaust using the relative wind direction data from the ship's meteorological instrumentation.

Flasks for $\delta(O_2/N_2)$ were taken in triplicate and connected in series downstream of a pump. The pump body and valve plates were aluminum, and the structured diaphragms were made of PTFE. When in use the flow rate (3.2 L min$^{-1}$) was higher than the in situ analyzer flow rates (100–200 mL min$^{-1}$). Air was dried with a magnesium perchlorate. During sampling, the line was flushed for 5 minutes before any air was directed to the flasks, then a bypass valve was opened and the flasks were flushed

for an additional 15 minutes before they were sealed again. After closure, the pressure of the flask was about 1.6 bars. Flasks were analyzed with a mass spectrometer at MPI-BGC (Brand, 2005). Storage times were less than two months.

Dissolved gas measurements were carried out by means of an autonomous setup for along-track measurements of $CO_2$, $N_2O$ and CO, which combined the analytical approaches from Pierrot et al. (2009) and Arévalo-Martínez et al. (2013). A full description can be found in Arévalo-Martínez et al. (2019).

The estimated uncertainty of the dissolved $CO_2$ measurements was $\pm$ 2 $\mu$atm; of dissolved $O_2$ measurements, $\pm$ 4 $\mu$mol L$^{-1}$; of dissolved $N_2O$, $\pm$ 0.1 nmol L$^{-1}$. The uncertainty of the atmospheric measurements of $N_2O$ was $\pm$ 0.9 ppb.

## 2.8 Shipboard Air–Sea Flux Density Estimates

In situ oceanographic and meteorological data were taken from the *Meteor*'s instrumentation. In order to determine the total dissolved inorganic carbon (DIC) content of surface waters, total alkalinity was estimated from temperature, salinity, and

dissolved $O_2$ data, using the locally interpolated alkalinity regression (LIAR) version 2 (Carter et al., 2018). The dissociation constants of carbonic acid were also determined from temperature and salinity using the formulations of Millero et al. (2006). The total DIC content was then estimated from the total alkalinity and $f CO_2$. Meteorological data (air temperature, barometric

pressure, wind speed, *etc.*) was observed at a height of 37 m above sea level. The absolute wind speed measured on the *Meteor* was converted to $U_{10}$ through the relationship (Justus and Mikhail, 1976):

$$U_{10} = U_{meas} \left( \frac{z_{10}}{z_{meas}} \right)^n \tag{7}$$

$$n = \frac{0.37 - 0.0081 \cdot \ln(U_{meas})}{1 - 0.0881 \cdot \ln(\frac{z_{meas}}{10})} \tag{8}$$

where $U_{meas}$ is the wind speed in m s$^{-1}$, measured at some height $z_{meas}$, in m.

Marine surface flux densities of $CO_2$, $O_2$, and $N_2O$ were estimated for the vessel location throughout M99 from shipboard measurements of atmospheric dry air mole fractions and dissolved aqueous concentrations, according to Equation 9. While flux densities of CO were also determined, they are not discussed in this manuscript as the shipboard-measured flux was too small to be detected as an atmospheric anomaly at NDAO (contrary to what was reported in Morgan (2015)). In the case of

$O_2$, the atmospheric concentration was measured only sporadically with flask samples, so it was taken as a static value, viz. the mole fraction of $O_2$ in standard dry air, 0.2093 (Aoki et al., 2019). The in situ aqueous solubility of $O_2$ was calculated using the equations of García and Gordon (1992), of $N_2O$ using those in Weiss and Price (1980), and of $CO_2$ using Weiss (1974). Sea-to-air fluxes (net evasion) are positive.

The flux density ($F$, in units of mol m$^{-2}$ s$^{-1}$) is typically determined according to (Garbe et al., 2014):

$$F = k_w(C_w - \alpha C_a) \tag{9}$$

where $k_w$ is the gas transfer (or piston) velocity, in m s$^{-1}$, $C_w$ is the dissolved concentration in the water phase (mol m$^{-3}$), and $C_a$ is the concentration of the species in the air in the same units. The formulation can also be altered to accommodate units of partial pressure in both phases. The expression $\alpha C_a$ gives the dissolved concentration in the water phase directly at the surface; $\alpha$ is the Ostwald solubility coefficient: the reciprocal of the dimensionless air–water partition coefficient ($K_{AW}$) for

some temperature, $T$, and salinity, $S$ (Mackay and Shiu, 1981).

As there is no definitive $k_w$–$U_{10}$ parameterization, fluxes were computed with two different parameterizations of $k_w$: that of Wanninkhof (2014) ($k_{W14}$) and McGillis et al. (2001) ($k_{McG01}$):

$$k_{W14} = 0.251 U_{10}^2 \left( \frac{Sc}{660} \right)^{-0.5} \tag{10}$$

$$k_{McG01} = (3.3 + 0.026 U_{10}^3) \left( \frac{Sc}{660} \right)^{-0.5} \tag{11}$$

In these equations, $U_{10}$ is the wind speed at 10 m's height, and Sc is the Schmidt number of a particular gas at in situ conditions (Jähne et al., 1987; Wanninkhof, 1992). The Schmidt number is dimensionless, and $U_{10}$ is in units of m s$^{-1}$, producing $k_w$

in units of cm hr$^{-1}$. The Schmidt number is scaled to the reference conditions of the parameterization; Schmidt numbers were calculated at in situ conditions by dividing the kinematic viscosity of seawater by the diffusivity of a given gas, using an Eyring-style equation (Eyring, 1936; Jähne et al., 1987):

$$D = A \exp\left(\frac{E_a}{RT}\right) \tag{12}$$

where R is the ideal gas constant, T is temperature, and $E_a$ is the activation energy for diffusion in water. $A$ and $E_a$ are determined through fits to experimental data. For $O_2$ these were taken from Ferrell and Himmelblau (1967), for $CO_2$ from Jähne et al. (1987), and for $N_2O$ from Bange et al. (2001). A salinity correction of 4.9% per 35.5 psu was then applied, this number being the average decrease in diffusivity seen by Jähne et al. (1987) for He and $H_2$ in an experiment involving artificial seawater. The kinematic viscosity of seawater at in situ conditions was determined by calculating the dynamic viscosity after Laliberté (2007) and the density after Millero and Huang (2009).

## 3 Results and Discussion

### 3.1 Upwelling Events

Out of 741 days, 219 days met the SST criteria, but only 173 of these met both the SST and WS criteria, representing 23% of the two-year study period. From these 173 days with upwelling events, 102 had atmospheric conditions that allowed for the detection of an anomaly in the atmospheric time series. 21 of excluded events were filtered based on HYSPLIT back-trajectories; the rest were excluded based on local meteorology or elevated CO. This was a conservative approach, since the filtering criteria in part relied on HYSPLIT model results, which could misrepresent the transport pathway and lead to the exclusion of an upwelling event that in fact was favorable for carrying an air mass influenced by upwelling to the station. Despite the greater prevalence of equatorward winds during austral summer, the distribution of events displayed little seasonality (Figure 3), reflecting the fact that upwelling is a short-term, intraseasonal phenomenon, forced by specific atmospheric conditions (Risien et al., 2004; Goubanova et al., 2013). While upwelling in this area is perennial, seasonality is seen in the intensity of upwelling due to the annual migration of the South Atlantic Anticyclone, with a minimum in austral winter (Hagen et al., 2001; Hardman-Mountford et al., 2003; Peard, 2007; Veitch et al., 2009; Hutchings et al., 2009).

An example of an upwelling event is given in Figures 1, 4, and 5. On November 27[th], 2013, high winds resulted in the creation of a very large pool of colder water on the surface that persisted for four days, until winds relaxed and upwelling temporarily ceased until the 4[th] of December, when SST dropped again. During these two low SST pulses, chl-*a* values were higher. A change in the background values of atmospheric potential oxygen (APO), $N_2O$, and $CH_4$ was seen, with a smaller anomaly for $CO_2$. The largest anomalies for each species came when high winds were from the coastal sector on November 28[th].

If the area of high flux was close to the coast, anomalies could arrive within a few hours at NDAO, with the development of the afternoon sea breeze. If the region of flux was closer to Lüderitz, the arrival time could be delayed by as much as 50 hours,

depending on the wind speed and the degree of meandering of the air mass. Back-trajectories implied that despite the high wind speeds usually seen in this coastal zone, significant travel time (1 to 2 days) could be expected for most air masses of interest. As a result, the marine surface flux associated with an atmospheric anomaly could only be said to have taken place with 50 hours of its detection. The magnitudes of the average atmospheric anomalies and their corresponding flux density estimates are given in Table 1.

## 3.2 Estimated Top-Down Flux Densities

$CO_2$ fluxes were positive for all upwelling events, with an average flux density of $0.70 \pm 0.42$ $\mu$mol m$^{-2}$ s$^{-1}$ and a maximum value of 2.0 $\mu$mol m$^{-2}$ s$^{-1}$. During upwelling conditions, it is not uncommon to see such strong outgassing of $CO_2$ in a coastal upwelling region; the biological response to new nutrients takes some days to draw down DIC levels (Torres et al., 1999; Loucaides et al., 2012; Cao et al., 2014). Santana-Casiano et al. (2009) used underway systems on cargo ships and weekly wind speeds to arrive at a mean flux between ca. $-0.06$ and 0.03 $\mu$mol m$^{-2}$ s$^{-1}$ for the Lüderitz region, with peak rates as high as 0.06 $\mu$mol m$^{-2}$ s$^{-1}$ in August. González-Dávila et al. (2009) found that the Lüderitz region is under-saturated with respect to $CO_2$, with only upwelled waters seeing oversaturation, with average fluxes on the order of $-0.03 \pm 0.3$ $\mu$mol m$^{-2}$ s$^{-1}$. The flux densities for $CO_2$ reported in this study are not necessarily in conflict with these studies, since a yearly-averaged flux density is a different quantity from the event-based flux densities. The implication instead is that flux densities attributed to upwelling events are high enough to contribute significantly to the carbon balance of the Lüderitz and Walvis Bay regions.

Typical $O_2$ flux densities were about $-5$ $\mu$mol m$^{-2}$ s$^{-1}$. The $O_2$ flux densities inferred from APO are preferred over those calculated with atmospheric $\delta(O_2/N_2)$, as APO is explicitly formulated to remove land biosphere influences on the relative change in oxygen abundance. A linear regression between the two estimated flux densities yielded a slope of 0.91 and a coefficient of determination of $R^2 = 0.98$. The estimated APO flux density was 11% lower on average than the $\delta(O_2/N_2)$-inferred flux density. That the flux densities for $O_2$ calculated with these two formulations are close is a good indication that the atmospheric data was primarily influenced by marine fluxes. These are high flux densities for the marine environment; the estimated average flux density for the entire mid-South Atlantic was $\sim$0.03 $\mu$mol m$^{-2}$ s$^{-1}$ in the inverse modeling study of Gruber et al. (2001), and ca. 0.06 $\mu$mol m$^{-2}$ s$^{-1}$ in the forward run of a coupled climate and ocean biogeochemistry model of Bopp et al. (2002).

The average flux density attributable to specific upwelling events for $N_2O$ was $0.66 \pm 0.4$ nmol m$^{-2}$ s$^{-1}$, moderate for a coastal upwelling system. Flux densities ranged from 0.091–1.3 nmol m$^{-2}$ s$^{-1}$. From surface data from cruise 258 of the RV *Africana* (Emeis et al., 2018) in the NBUS we estimate a maximum flux density of about 0.07–0.5 nmol m$^{-2}$ s$^{-1}$. Frame et al. (2014) observed flux rates as high as 0.52 nmol m$^{-2}$ s$^{-1}$ in the Cape Frio upwelling cell. The 3-D coupled physical/biogeochemical model of Gutknecht et al. (2013b, a) predicts an 8-year mean flux density of 0.02–0.16 nmol m$^{-2}$ s$^{-1}$ for the Walvis Bay region, including both shelf and deeper waters as far west as 10° E. The mean flux density of the entire M99 cruise was $\sim$0.03 nmol m$^{-2}$ s$^{-1}$. The model predicts a maximum flux density at the coast (22–24° S) of 0.6 nmol m$^{-2}$ s$^{-1}$. Arévalo-Martínez et al. (2019) reported higher $N_2O$ flux densities from a domain encompassing the latitude range of

16–28.5°S, ranging between 0.03 and 1.67 nmol m$^{-2}$ s$^{-1}$. For the Lüderitz/Walvis Bay area they found values similar to this study: up to 0.20 nmol m$^{-2}$ s$^{-1}$.

The moderate, positive $CH_4$ spikes (∼15 ppb) seen during upwelling events corresponded to an average flux density of about 5 nmol m$^{-2}$ s$^{-1}$ and a maximum of 36.4 nmol m$^{-2}$ s$^{-1}$, a high value even for coastal waters. There are few reported measurements of flux densities or dissolved $CH_4$ for the Benguela region to place these estimates in context. In what are likely the first measurements of dissolved methane in the Benguela, Scranton and Farrington (1977) observed concentrations near Walvis Bay at multiple depths in the 200–900 nM range. The only other available data known to the authors are from cruise 258 of the RV *Africana* in 2009. These measurements were taken at a variety of depths (up to 400 m) and ranged from 4.7 to 140.0 nM. Using the nine samples taken from the top 15 m from this cruise, at in situ conditions, a flux of 0.08–1.5 nmol m$^{-2}$ s$^{-1}$ would be expected, from dissolved concentrations ranging from 6.0 to 140 nM. Naqvi et al. (2010) used the data from Scranton and Farrington (1977) and from Monteiro et al. (2006) to estimate flux densities of 0.03–8.7 nmol m$^{-2}$ s$^{-1}$, the upper end-member of which compares favorably with the rates found in this study for the same ocean region. Though the latter study observed concentrations at a mooring near Walvis Bay as high as 10 $\mu$M, these data were from an uncalibrated probe and are not usable for a direct comparison to other campaigns. In other EBUS, reported flux densities are much lower.

### 3.3 M99: Atmospheric Measurements

Only $CO_2$ and $CH_4$ were measured continuously in the atmosphere on M99. During the first days of the cruise, before the main upwelling event, a synoptic event brought elevated mixing ratios of $CO_2$ and $CH_4$ offshore (Figure 6). The shipboard measurements of $CO_2$ showed sporadic enhancements relative to the smoothed station background. These enhancements coincided with the regions of higher flux closer to the coast encountered under upwelling conditions (Figure 6 and 7). In contrast, apart from the initial synoptic event, $CH_4$ was consistently at background levels, usually below the value seen at NDAO.

### 3.4 M99: Dissolved Gas Concentrations

All three species measured underway in the water phase showed the largest deviation from atmospheric equilibrium closest to shore (Figure 7). $f$CO$_2$ ranged from 355.5 to 852.3 $\mu$atm (87.8% to 207.3% saturation), with most of the observed oversaturation occurring under upwelling conditions. Dissolved oxygen was mostly at saturation or slightly above, although close to shore the concentration dropped to a minimum of 180.8 $\mu$M (67% saturation). $N_2O$ surface concentrations were rather low for an upwelling region, but agreed well with previously reported values, with the maximum observed concentration of 20.5 nM being comparable to the highest values seen by Frame et al. (2014) for surface measurements in the same region. Recently, Arévalo-Martínez et al. (2019) reported surface $N_2O$ concentrations ranging between 8 and 31 nM for the northern Benguela region. Besides this, the only other in situ measurements of $N_2O$ in the Benguela region known to the authors are in the Marine Methane and Nitrous Oxide (MEMENTO; https://memento.geomar.de/de) database (Bange et al., 2009; Kock and Bange, 2015), from cruise 258 in 2009 of the RV *Africana*. In this dataset, dissolved $N_2O$ concentrations for surface waters (the top 15 m) were in the range of 5–51 nM, which brackets the range measured during M99.

The main upwelling event of the cruise, in the Lüderitz/Walvis Bay cell, began on August 4th, 2013, and lasted until August 11th (Figure 8). Wind speeds declined rapidly after the 8th. The upwelling event was encountered by the *Meteor* starting on the 8th, as the vessel reached an upwelling filament, the outer edge of which was subject to net evasion of all three gases ($CO_2$, $O_2$, $N_2O$), likely a result of warming temperatures that would reduce their solubility. The positive flux ratio of $O_2:CO_2$ is generally only produced when thermal processes dominate the air-sea flux.

### 3.5 Comparison of Top-Down and Bottom-Up Flux Density Estimates

Due to local wind variability, suitable conditions for detecting the upwelling event encountered by the *Meteor* were only seen at NDAO on the 6th, 8th, and 10th of August, 2013. As the vessel was not always in the upwelling cell, only a single atmospheric anomaly at NDAO could be matched to in situ shipboard measurements, namely an anomaly occurring on August 10th. The highest flux rates (positive for $CO_2$ and $N_2O$, and negative for $O_2$) were seen within the recently upwelled waters experiencing high wind speeds. Fluxes displayed coupling between all three species, though the area of high flux density for $O_2$ and $N_2O$ was more sharply defined than for $CO_2$.

The top-down estimates agreed reasonably with the corresponding mean shipboard estimates for a selected 7.5 hour period that coincided with the largest flux densities (Figure 8 and Table 2), but overestimates the flux density relative to the mean shipboard estimate for the flux event. Agreement was best for $CO_2$, but was as poor as a factor of 3 for $N_2O$, though given the simplicity of the top-down estimate, this seems fairly reasonable.

The choice of gas transfer velocity parameterization ($k_w$) was significant in determining the magnitude of the agreement, and the degree of this correspondence between the top-down and shipboard gas transfer velocity-specific estimate varied between species. The cubic relationship of $k_{McG01}$ produced higher fluxes during the M99 event, and was in better agreement with the top-down estimate. While this provides a measure of confidence in the top-down flux density estimates, it should be noted that the neither of the estimated uncertainties for the top-down or bottom-up approaches account for errors incurred by the simplifying assumptions within their formulations, and these comparisons should not be considered a "calibration" of the flux determination or $k_w$ parameterization. We suspect that our top-down estimates are overestimates. Cubic relationships with wind speed, derived mostly from eddy covariance data, have not been disproved per se, but after reevaluation of field data (Landwehr et al., 2014), the weight of evidence seems to favor a quadratic relationship (Roobaert et al., 2018), and hence we prefer $k_{W14}$ for comparison.

### 3.6 Model Uncertainty and Limitations

The model makes many simplifying assumptions, which we have attempted to incorporate into our uncertainty assumptions. The assumption of a constant boundary layer height, $h$, is of course not realistic, but we have attempted to account for natural variations in this parameter with our uncertainty range. The uncertainty in the distance traveled depends on the accuracy of the back-trajectories, although major errors in transport (*e.g.* a continental air mass causing the anomaly) are likely to be excluded due to the filtering criteria of the atmospheric record. We also note that the model is sensitive to the dilution rate constant, $q$, and that this is the input we know the least well. Previous studies suggest that our in situ determination is reasonable; (Price

et al., 2004) used multiple species and a model approach to arrive at a mean $q$ of $0.010 \pm 0.004 \text{ hr}^{-1}$, which is nearly identical to our value of $0.011 \pm 0.006 \text{ hr}^{-1}$, although the spatial and temporal scale they considered was larger. Dillon et al. (2002) observed rates that were higher for a Sacramento pollution plume, ca. $0.2 \text{ hr}^{-1}$.

While we cannot determine for specific events when the simplifying assumptions of the model are problematic, we assume
that variations in the dilution rate, the boundary layer height, and errors in transport generally average out, and that the estimated average flux from upwelling events is representative. If we "tune" the model to match the M99 event to the $k_{W14}$ fluxes for each species, we get a lower value of $0.0058 \pm 0.007 \text{ hr}^{-1}$. Using this value of $q$ we obtain the tuned fluxes reported in Table 2. Though these fluxes have a higher uncertainty associated with them, due to the higher uncertainty in $q$, we recommend these values, as the top-down and bottom-up comparison suggests that the model overestimates the fluxes.

## 3.7 Stoichiometry

Correlation slopes of atmospheric species can provide further confidence and insight into source processes, if there is an underlying biogeochemical relationship. The well-known inverse relationship between $N_2O$ and $O_2$ in the ocean, for instance, is a result of organic matter decomposition and nitrification (Cohen and Gordon, 1979; Nevison et al., 2003; Naqvi et al., 2010; Frame et al., 2014). The tight coupling between $N_2O$ and $O_2$ seen in surface concentrations during M99 is preserved during
air–sea gas exchange, as these gases behave similarly (Figure 9). The observed linear correlation in surface waters between these two species is also an indication that intense denitrification was not occurring in the OMZ at the time of sampling, which if it were taking place, would lead to a breakdown in this relationship at low $O_2$ (Cohen and Gordon, 1979). The approximate molar ratio of these two species, $-0.8 \times 10^{-4}$ to $-1 \times 10^{-4}$ ($N_2O:O_2$; mol mol$^{-1}$), is the same observed by Lueker et al. (2003) for the Trinidad Head region, and appears to be a globally consistent value (Nevison et al., 2005; Manizza et al., 2012).

This linear regression slope is often expressed in terms of the excess $N_2O$ (measured $N_2O$ minus $N_2O$ at saturation) and apparent oxygen utilization (saturation minus observed), $\Delta N_2O$–AOU in nmol $\mu$mol$^{-1}$. Quantified in this way, it has been used as an estimate of the yield of $N_2O$ as a function of the amount of oxygen consumed (Nevison et al., 2003). However, the relationship is not strictly linear, since $N_2O$ production is enhanced at low oxygen levels (Nevison et al., 2003; Naqvi et al., 2010; Trimmer et al., 2016). $\Delta N_2O$–AOU is also sensitive to mixing, as $N_2O$ production rates vary widely in the
ocean, meaning that the mixing of water masses with different compositions can overwhelm the in situ production signal (Suntharalingam and Sarmiento, 2000; Nevison et al., 2003). The $\Delta N_2O$–AOU for M99 was $0.088 \pm 0.003$ nmol $\mu$mol$^{-1}$, with an intercept of $1.6 \pm 0.04$ nmol, and an $R^2$ of 0.69. This is a low value, nearly identical to results from the eastern basin of the sub-tropical North Atlantic, where South Atlantic Central Water is found (Cohen and Gordon, 1979; Suntharalingam and Sarmiento, 2000; Walter et al., 2006), and the eastern equatorial Atlantic (Arévalo-Martínez et al., 2017). Frame et al. (2014)
showed that most of the $N_2O$ in the Benguela Current region is produced in the water column and in the sediment by nitrifier denitrification (Frame et al., 2014). Nevertheless, a substantial portion of this $N_2O$ remains at depth (with a concentration maximum at 200–400 m) and is advected away from the region, without the chance for atmospheric release (Gutknecht et al., 2013b, a; Frame et al., 2014). Hence, the low $\Delta N_2O$–AOU value found for the *Meteor* cruise probably reflects both physical and biogeochemical dynamics.

In the case of variations of $O_2$ and $CO_2$, the stoichiometry of surface waters is not preserved after air–sea exchange, as the majority of carbon is speciated in the carbonate system, and only the portion that remains as dissolved $CO_2$ is available for air–sea gas exchange. This leads to a change in the ratio, for instance, from $0.58 \pm 0.03$ in surface waters to $-6.53 \pm 0.42$ in the atmosphere, for the upwelling event encountered during the R/V *Meteor* cruise M99 (Figure 8). These two species can become decoupled through the influences of changing solubility, which would drive evasion of both gases, and net biological production, which would drive evasion of $O_2$ and invasion of $CO_2$. These complicating influences are the likely reason for the poorer correlation seen between these two species when compared with $N_2O$ and $O_2$.

While the top-down flux density estimates cannot be confirmed with shipboard estimates of flux densities for $CH_4$, it is worth considering that concentrations of methane in bottom waters on the Namibian shelf are likely the highest ever measured in an open coastal system. Values as high as 475 $\mu$M in the bottom waters and greater than 5,000 $\mu$M in sediment porewaters have been observed (Scranton and Farrington, 1977; Monteiro et al., 2006; Brüchert et al., 2009; Naqvi et al., 2010). In the water column, the concentration maxima is usually at the seabed or in bottom water, but it is variable and can even occur at the surface (1 m) (Brüchert et al., 2009). Dissolved methane concentrations are tightly coupled with $O_2$ and show considerable variability, with elevated concentrations being triggered by episodes of hypoxia (Monteiro et al., 2006; Brüchert et al., 2009). The pulse-like nature of $CH_4$ in the Benguela means that the full range of dynamics cannot be captured with a campaign-based sampling approach (Brüchert et al., 2009). What is clear is that there is a tremendous amount of methane production at depth, but that the source is variable in strength (Emeis et al., 2004; Brüchert et al., 2009). In light of the fact that large pockets of free methane gas are contained in the sediment in the Walvis Bay region, as well as the existence of craters and pockmarks on the seafloor, combined with observation of bubble streams from the seabed, suggest a mechanism by which methane produced in sediments can be abruptly transported to the surface and hence, avoid oxidation (Emeis et al., 2004; Brüchert et al., 2006, 2009). Consequently, the Benguela Current is a suspected source of $CH_4$ to the atmosphere, but the amount is ill-constrained (Naqvi et al., 2010); Emeis et al. (2018) recently estimated the total annual emission for the entire system to be less than 0.17 Tg $CH_4$ yr$^{-1}$.

Interestingly, methane was not well-correlated with either APO or $CO_2$ in the atmosphere during all upwelling events, suggesting a spatial decoupling (since a cross-correlation analysis indicated this was not a result of lag/temporal decoupling) between methane and these two species. While background observations of $CH_4$ were generally well-correlated with $CO_2$ and $O_2$ at NDAO, only some upwelling events showed such coupling; it seems there is a general relationship between methane and oxygen, but it is not consistent and is occasionally non-existent. Unfortunately, since there are still very few measurements of water-column $CH_4$ in the Benguela, a full explanation of the methane source remains elusive. From these atmospheric trends it can only be deduced that there is some separate biogeochemical influence on methane that is not exerted over $CO_2$, $O_2$, or $N_2O$. This observation is arguably consistent with the concept of a dominant sedimentary source of methane that is more localized within the inshore mud belt, where high POC fluxes have created a thick layer of diatomaceous ooze containing free methane gas pockets (Emeis et al., 2004; Brüchert et al., 2006; van der Plas et al., 2007; Brüchert et al., 2009).

## 3.8 Regional Flux Context and Future Steps

To put our estimated flux densities into the context of regional greenhouse gas budgets, it is necessary to have an approximation of the area involved. For the sake of discussion only, we can very roughly estimate the total annual flux of each species from the Walvis Bay/Lüderitz domain by assuming that grid cells with wind speeds above 3.5 m s$^{-1}$ and a deseasonalized SST below $-0.1°$ C were grid cells with upwelling, that upwelling extends to the coast, and that each grid cell with upwelling had the same flux density, *i.e.* the top-down estimated flux density for a given upwelling event. This approach is very crude, and would result in an underestimate, since transport conditions were not always conducive to observing an upwelling event. This would result in annual fluxes of $\geq 206 \pm 151$ Gmol yr$^{-1}$ for $CO_2$, $\geq -1.6 \pm 1.1$ Tmol yr$^{-1}$ for $O_2$, $\geq 272 \pm 248$ Mmol yr$^{-1}$ for $N_2O$, and $\geq 3.3 \pm 2.8$ Gmol yr$^{-1}$ for $CH_4$ from upwelling events.

Such an estimate for $CO_2$ is substantial when compared to the net flux that Laruelle et al. (2014) estimated of $-424.9$ Gmol yr$^{-1}$ for the entire Benguela region, or the $-141.5$ Gmol yr$^{-1}$ found by Gregor and Monteiro (2013) for the southern Benguela. A high flux of $CO_2$ is possible due to the higher wind speeds and the more remineralized character of the South Atlantic Central Water that upwells at Lüderitz. Using inverse methods, Gruber et al. (2001) constrained the net flux of oxygen for the temperate South Atlantic (an area of $1.5 \times 10^7$ km$^2$) to be 15.5 Tmol yr$^{-1}$, which suggests that upwelling events in the Lüderitz region could be regionally significant. An annual source of this magnitude for $N_2O$ is modest when compared to the budgetary calculations of other coastal regions. As the global coastal upwelling source of $N_2O$ is estimated to be 7,140 Mmol yr$^{-1}$ (Nevison et al., 2004), the emissions for the Lüderitz/Walvis Bay region would represent 3.2% of the area but only 1.7% of these emissions. For $CH_4$ this approximation is two to three times higher than the net evasion from the Arabian Sea (Bange et al., 1998), and 10 to 20 times greater than the annual release of $CH_4$ from the entire Mauritanian upwelling system (Kock et al., 2008; Brown et al., 2014), but modest when compared to the estimate of Monteiro (2010) for the entire Bengulea Upwelling System, 56 Gmol yr$^{-1}$, or the 10 Gmol yr$^{-1}$ proposed by Emeis et al. (2018) for the NBUS.

A full regional top-down quantification of the annual fluxes–as opposed to the event-based averages we present here—of the Benguela from land-based atmospheric measurements would require additional efforts, such as a Bayesian synthesis inversion of the data, where a prior surface flux field is adjusted to best match the atmospheric observations. This approach relies on an accurate knowledge of atmospheric transport, and ideally a dense network of measurement sites. Currently there are only two such sites in the Benguela region known to the authors, NDAO and Cape Point Observatory (CPT) in South Africa. Until the observational coverage is improved, uncertainties will remain large. Given the cost and manpower involved in measuring fluxes in situ with a research vessel, it seems advantageous to pursue a denser network of regional sites along the southwestern coast of southern Africa. Such data would be of use to ongoing efforts to accurately determine terrestrial and marine greenhouse gas budgets for the region (*e.g.* Valentini et al., 2014; Bange et al., 2019).

In other EBUS, monitoring of some or all of atmospheric $CO_2$, $CH_4$, $N_2O$, and $\delta(O_2/N_2)$ is ongoing. Most of these sites are flask sampling locations, which are not as suitable as continuous measurements for capturing the short-lived nature of upwelling events. The California Current System is probably the best sampled, through two Advanced Global Atmospheric Gases Experiment (AGAGE) site, and numerous NOAA/ESRL activities. Long-term measurements are also made at the Cape

Verde Atmospheric Observatory off of the Mauritanian upwelling region. While these efforts are extremely valuable, a greater density of continuous atmospheric measurement sites could provide great benefit in quantifying greenhouse gas fluxes from upwelling systems.

## 4    Summary and Conclusions

We have shown that coastal atmospheric anomalies of $CO_2$, $O_2$, $N_2O$, and $CH_4$ can be related to upwelling events in the Lüderitz and Walvis Bay upwelling cells. Using a simple model, we have estimated the flux density of the four species during upwelling events. These top-down estimates of surface fluxes have been found to be in reasonable agreement with in situ surface flux densities as determined from shipboard measurements, although the top-down method underestimates the flux density before the model was tuned. Our study highlights the usefulness of continuous monitoring in order to achieve a more accurate estimation of the emissions of GHGs from coastal upwelling systems, since the large impact of small temporal and spatial scales of variability are prone to be overlooked by in-situ surveys.

The Lüderitz and Walvis Bay upwelling cells have been shown to be potentially an unusually large source of $CH_4$ to the atmosphere for the marine environment. In contrast, the region is a weaker source of $N_2O$, compared to other upwelling regions, a fact which has been predicted from modeling studies and noted in observations of dissolved concentrations and air–sea fluxes. This upwelling area also functions as a significant source term in the $CO_2$ budget of the Benguela Current.

We have focused here on upwelling events, because they are distinguishable from other sources of intraseasonal variability in the atmospheric record. A full top-down accounting of the greenhouse gas budget of the the Benguela could be accomplished through a Bayesian atmospheric inversion of one or more coastal stations. Based on our results and previous studies (Nevison et al., 2004; Lueker et al., 2003) we suggest to establish a network of high-resolution atmospheric GHG measurements adjacent to major coastal upwelling regions in order to monitor their GHG emissions. Continuous land-based monitoring will help to establish time series of coastal GHG fluxes which are needed to account for the seasonal variability and to detect both short-term and long-term trends of both marine and terrestrial coastal GHG emissions.

*Data availability.* Atmospheric data from NDAO is included as part of the Supporting Information. M99 $N_2O$ and $CO_2$ data are available from the MarinE MethanE and NiTrous Oxide database (MEMENTO, https://memento.geomar.de/) and the Surface Ocean $CO_2$ Atlas (SOCAT, https://www.socat.info/), respectively.

*Author contributions.* M.H. and J.L. conceived of the location and motivation for the atmospheric observatory. E.J.M., J.L., and T. Seifert conducted the atmospheric observations in Namibia of all species and on the RV *Meteor* for $CO_2$, $CH_4$, and $\delta(O_2/N_2)$. D.L.A.-M. and T. Steinhoff made the underway measurements of the other dissolved and atmospheric gases on the *Meteor*. E.J.M. wrote the manuscript, and all authors made substantial contributions to the text and/or analyses.

*Competing interests.* The authors declare that they have no conflict of interest.

*Acknowledgements.* The authors wish to express their gratitude to the Master, crew, and Chief Scientist (Detlef Quadfasel) of the RV *Meteor* during leg M99, and to the government of Namibia for permission to work in Namibian territorial waters. This work is a contribution to SOPRAN III (BMBF grant# FKZ 03F0662). SATRE collaborators Gregor Rehder and Jan Werner (Leibniz Institute for Baltic Sea Research)
5   operated the $N_2O/CO/CO_2$ underway system throughout the cruise. Tim Rixen (Leibniz Center for Tropical Marine Ecology/University of Hamburg) kindly provided the *FRS Africana* cruise data, as well as helpful comments on this manuscript. We also thank and acknowledge Gillian Maggs-Kölling, Theo Wassenaar, Jessica Sack, Tayler Chicoine, Robert Logan, and the Gobabeb community for their technical support and hospitality. The authors are grateful for the efforts and expertise of Armin Jordan, Willi Brand, Michael Hielscher, Bert Steinberg, Johannes Schwarz, and Jürgen Richter (MPI-BGC) in preparing and analyzing flask samples and gas cylinders. Author D.L.A.M. was
10  supported by the EU FP7 project InGOS (Grant Agreement #284274). Author E.J.M. was a part of the International Max Planck Research School for Global Biogeochemical Cycles when this work was conducted and acknowledges its funding and support. Funding for the research activities presented here was provided by the Max Planck Society.

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

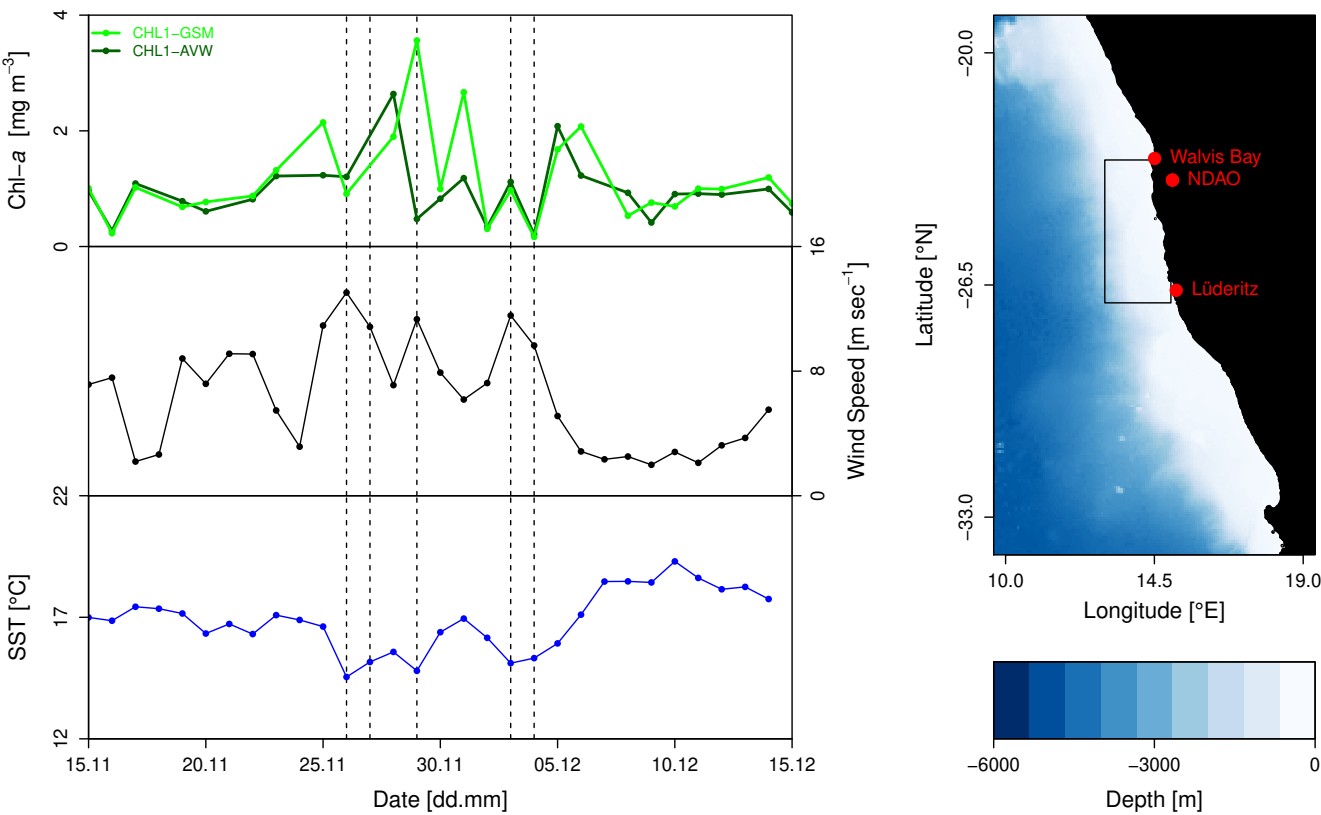

**Figure 1.** An example of an upwelling event at the end of 2013. The median chlorophyll *a* of the domain is shown over a period of one month, along with the domain-averaged 10-m wind speed and sea surface temperature (*left panel*). Days flagged as upwelling events are shown as dashed vertical lines. The Lüderitz/Walvis Bay domain is shown, overlain on a bathymetric map (*right panel*). Data is from Amante and Eakins (2009).

**Table 1.** Means of all atmospheric anomalies and top-down flux density estimates for identified upwelling events

| Quantity | $CO_2$ | $\delta(O_2/N_2)$, $O_2$ | APO, $O_2$ | $N_2O$ | $CH_4$ |
|---|---|---|---|---|---|
| | ($\mu$mol m$^{-2}$ s$^{-1}$) | ($\mu$mol m$^{-2}$ s$^{-1}$) | ($\mu$mol m$^{-2}$ s$^{-1}$) | (nmol m$^{-2}$ s$^{-1}$) | (nmol m$^{-2}$ s$^{-1}$) |
| Mean Anomaly $\pm 1\sigma$* | $2.0 \pm 1.1$ ppm | $-84 \pm 22$ per meg | $-75 \pm 18$ per meg | $1.9 \pm 0.9$ ppb | $14.9 \pm 17$ ppb |
| Mean Flux Density | 0.70 | $-6.1$ | $-5.4$ | 0.66 | 5.2 |
| Standard Deviation of Flux Density | 0.42 | 3.0 | 2.6 | 0.52 | 6.7 |
| Mean Uncertainty of Flux Density | 0.43 | 3.8 | 3.3 | 0.41 | 3.2 |
| Mean Tuned Flux Density | 0.65 | $-5.7$ | $-5.1$ | 0.61 | 4.8 |
| Standard Deviation of Tuned Flux Density | 0.40 | 2.9 | 2.5 | 0.50 | 6.3 |
| Mean Uncertainty of Tuned Flux Density | 0.7 | 6.0 | 5.4 | 0.7 | 5.1 |

*Flux density units in column header correspond to all values except for those in this row.

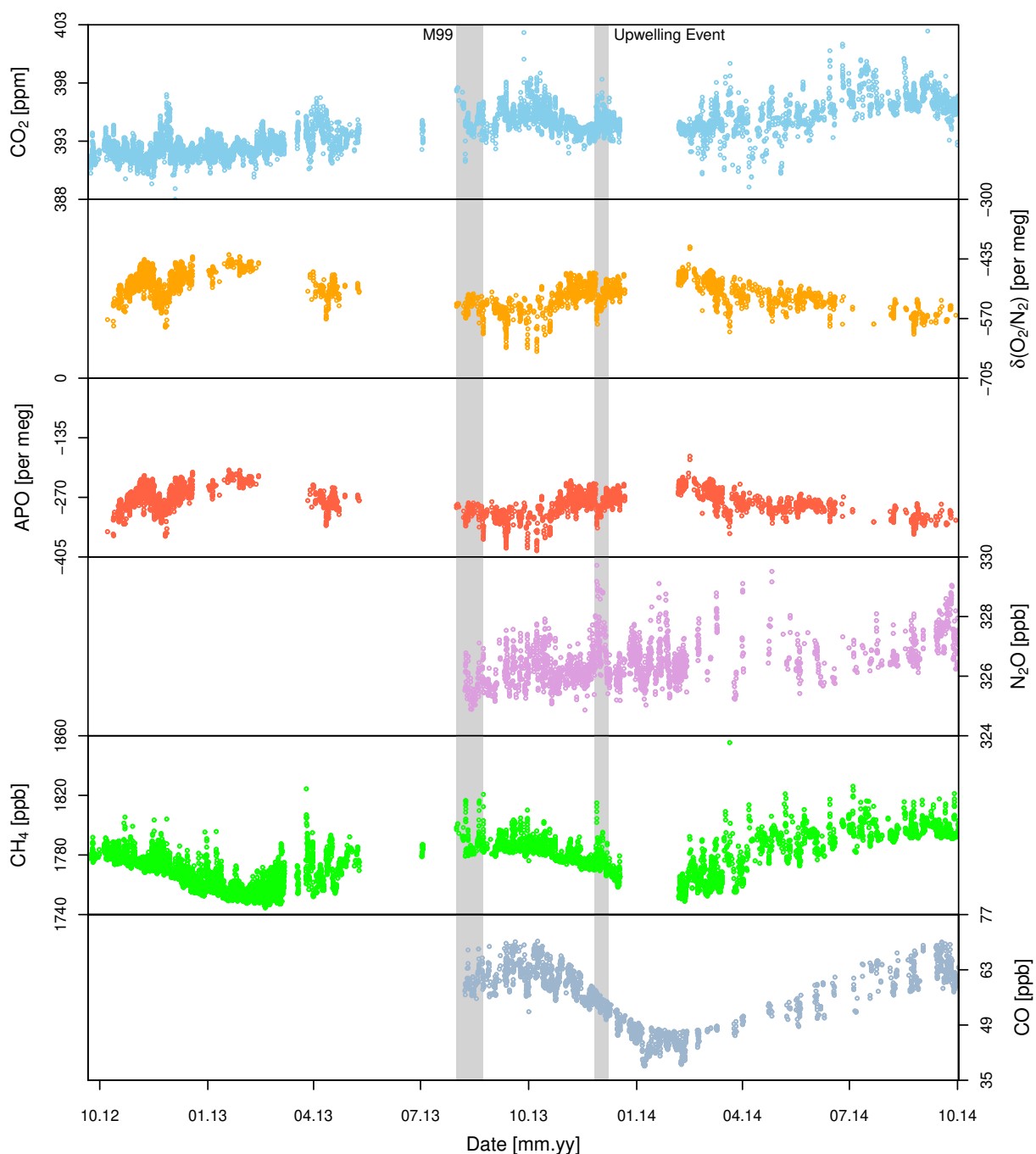

**Figure 2.** The NDAO time series, filtered based on back-trajectory and CO. The main measurands of the station are each plotted as 30 minute averages. The duration of M99 and the upwelling event discussed in the main text are both demarcated with gray rectangles and denoted with a label. The presence of an upper bound in the CO data is due to its use as a filtering criterion.

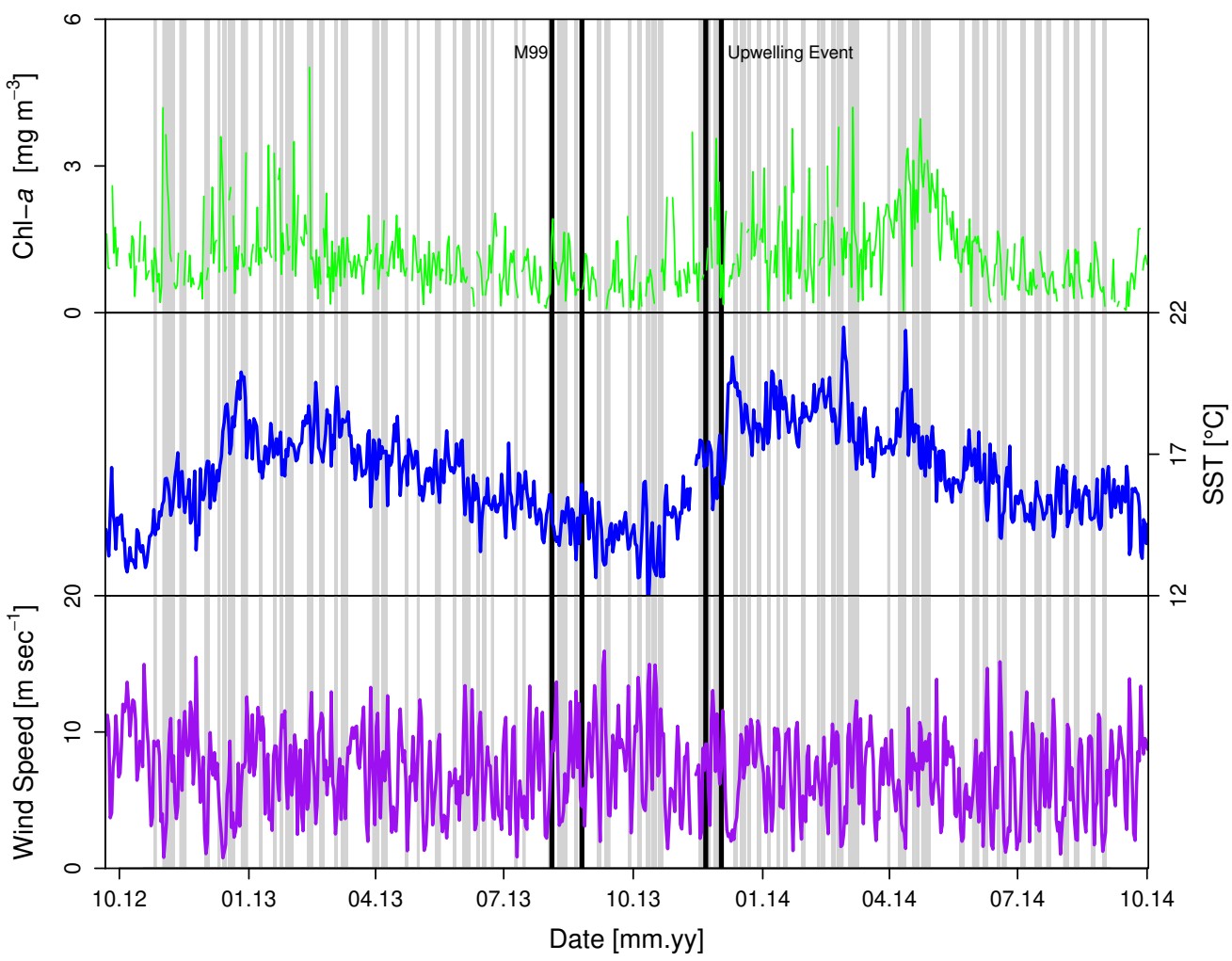

**Figure 3.** Surface chlorophyll *a*, temperature, and 10-m wind speed for the Lüderitz domain over the course of the two-year study period. Days which have been flagged as containing an upwelling event have been shaded. The duration of M99 and the upwelling event discussed in the main text are both demarcated with black rectangles and denoted with a label.

**Table 2.** Comparisons of top-down and underway flux density estimates for the M99 upwelling event (Aug 9 2013)

| Species | Unit | Top-Down | Shipboard ($k_{W14}$) | Shipboard ($k_{McG01}$) |
|---------|------|----------|----------------------|------------------------|
| $CO_2$ | $\mu$mol m$^{-2}$ s$^{-1}$ | $0.52 \pm 0.3$ | $0.45 \pm 0.2$ | $0.67 \pm 0.2$ |
| $\delta(O_2/N_2)$, $O_2$ | $\mu$mol m$^{-2}$ s$^{-1}$ | $-3.4 \pm 2$ | $-1.6 \pm 0.5$ | $-2.3 \pm 0.7$ |
| APO, $O_2$ | $\mu$mol m$^{-2}$ s$^{-1}$ | $-2.8 \pm 2$ | $-1.6 \pm 0.5$ | $-2.3 \pm 0.7$ |
| $N_2O$ | nmol m$^{-2}$ s$^{-1}$ | $0.42 \pm 0.2$ | $0.15 \pm 0.08$ | $0.22 \pm 0.1$ |

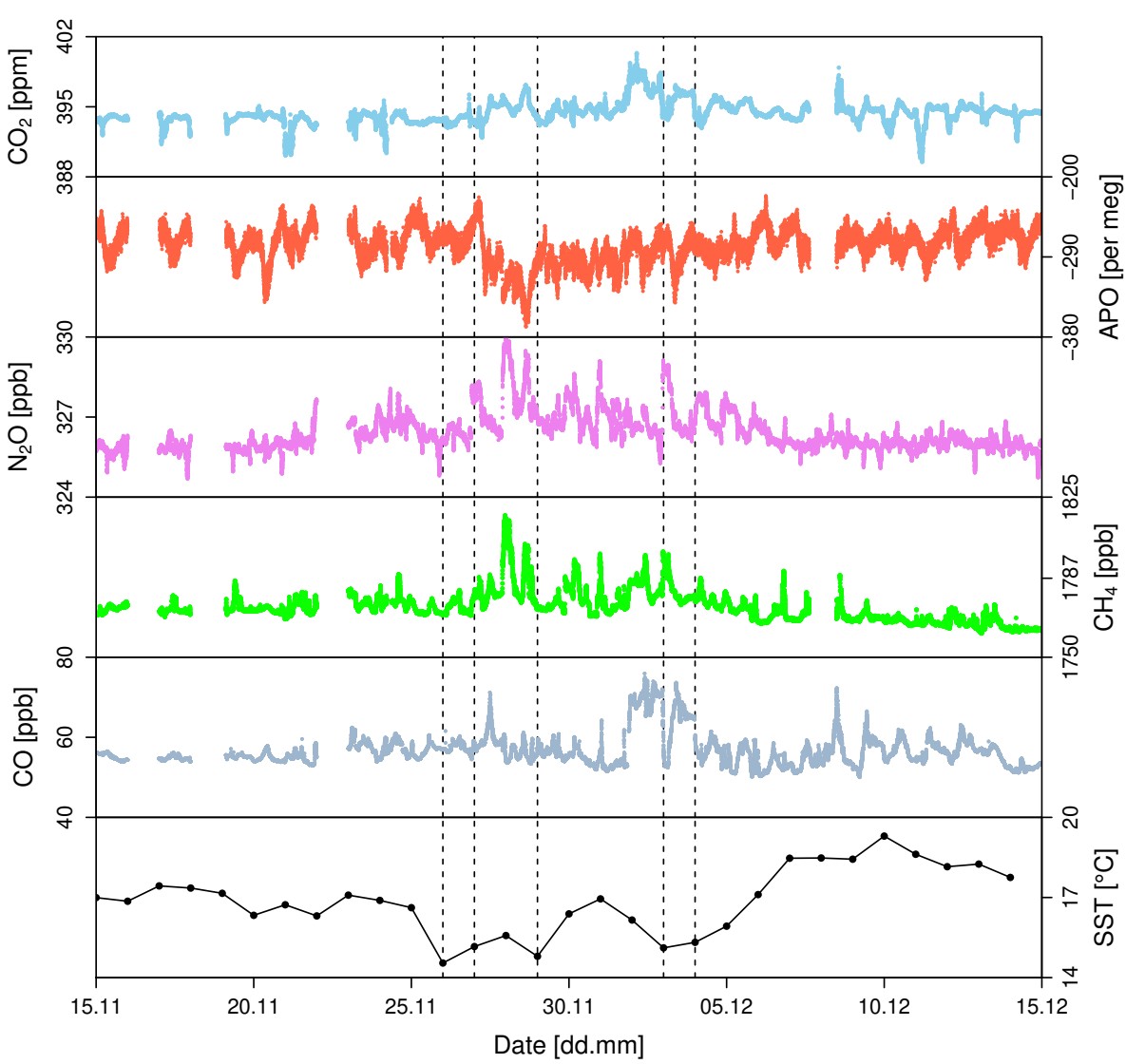

**Figure 4.** Atmospheric time series at NDAO throughout the upwelling event displayed in Figure 1. Days flagged as upwelling events are shown as dashed vertical lines.

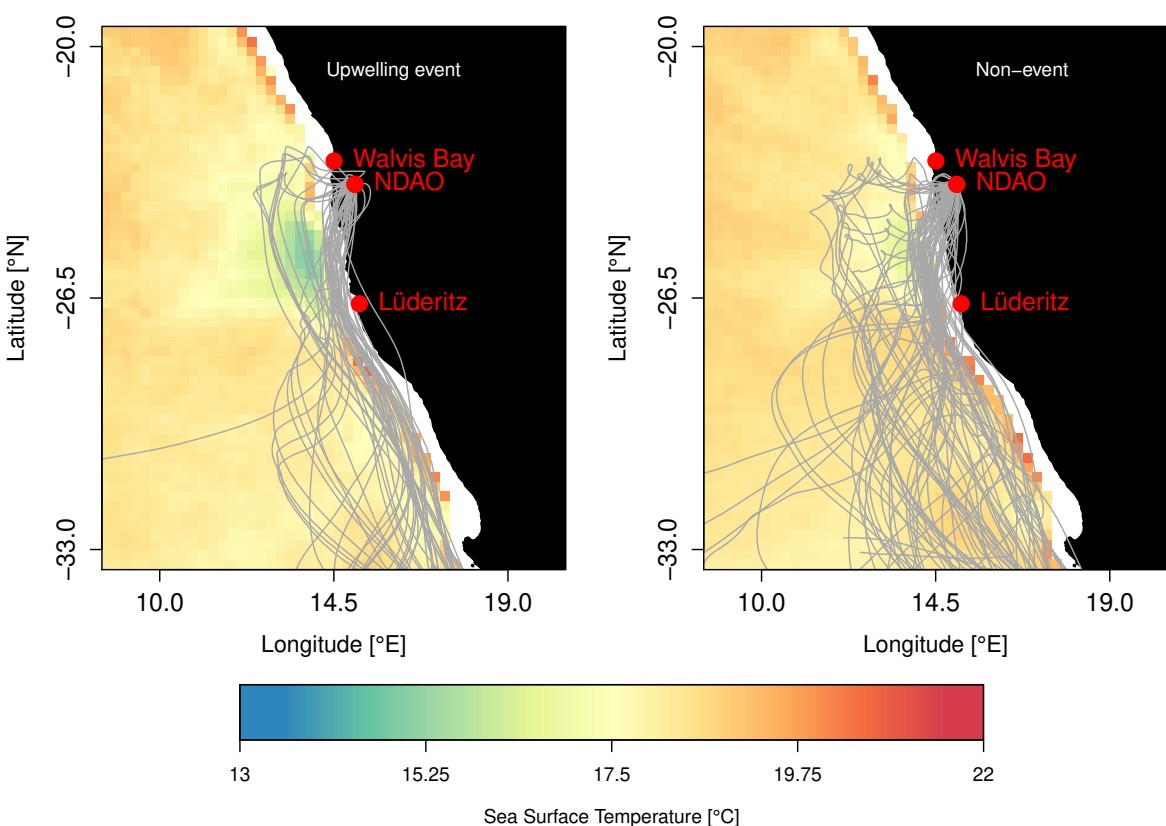

**Figure 5.** SST during (*left panels*) and preceding and after (*right panels*) the upwelling event described in Figure 2. Five-day back-trajectories calculated for the respective periods are overlain.

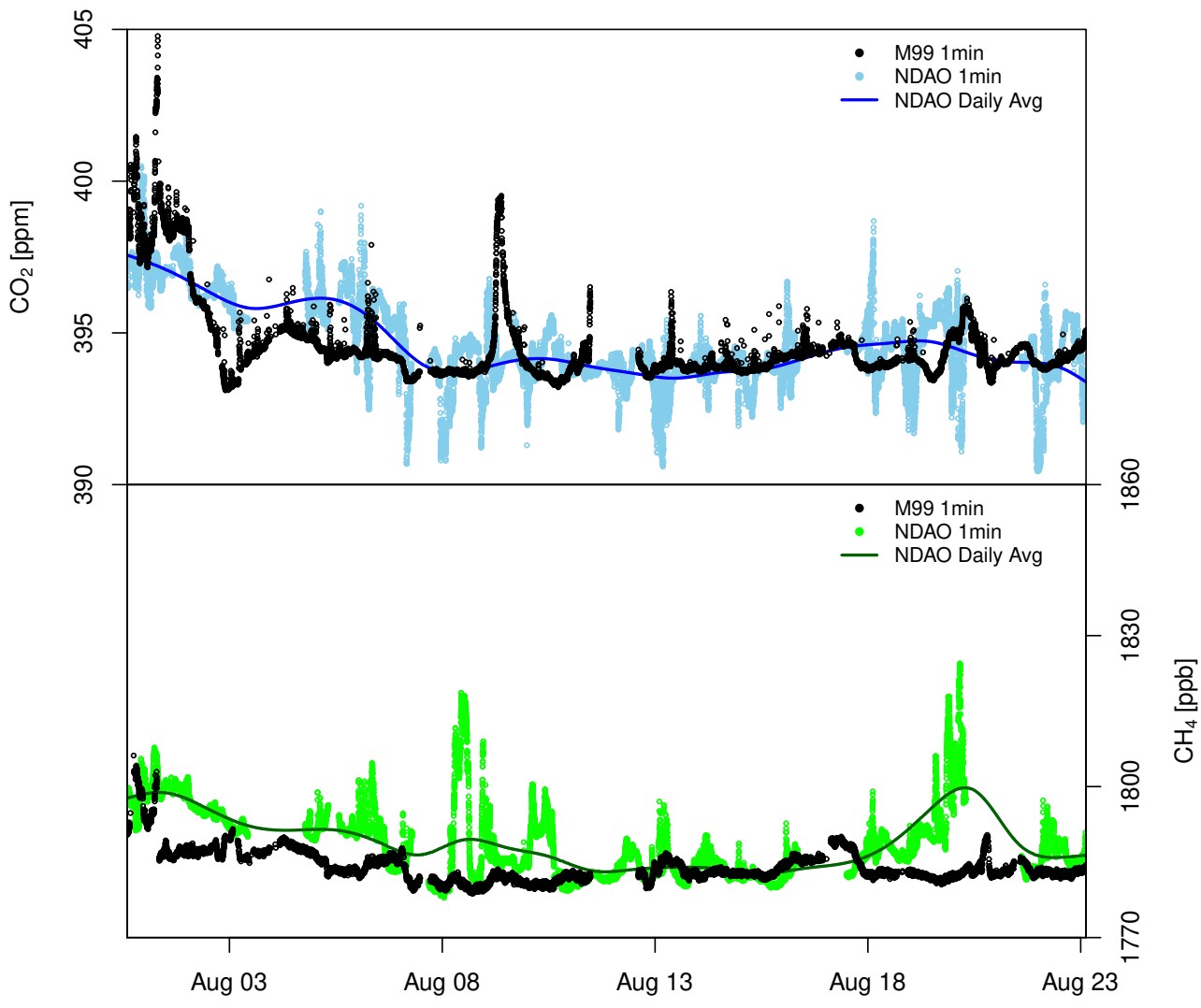

**Figure 6.** Atmospheric observations of $CO_2$ and $CH_4$ during the M99 cruise. Black points are the shipboard measurements and colored points are the NDAO measurements. Also shown for both species is a smooth fit to a rolling daily average. The diurnal variability was caused by a sea breeze/land breeze dynamic, with lower $CO_2$ mole fractions and higher $CH_4$ mole fractions occurring at night during the land breeze.

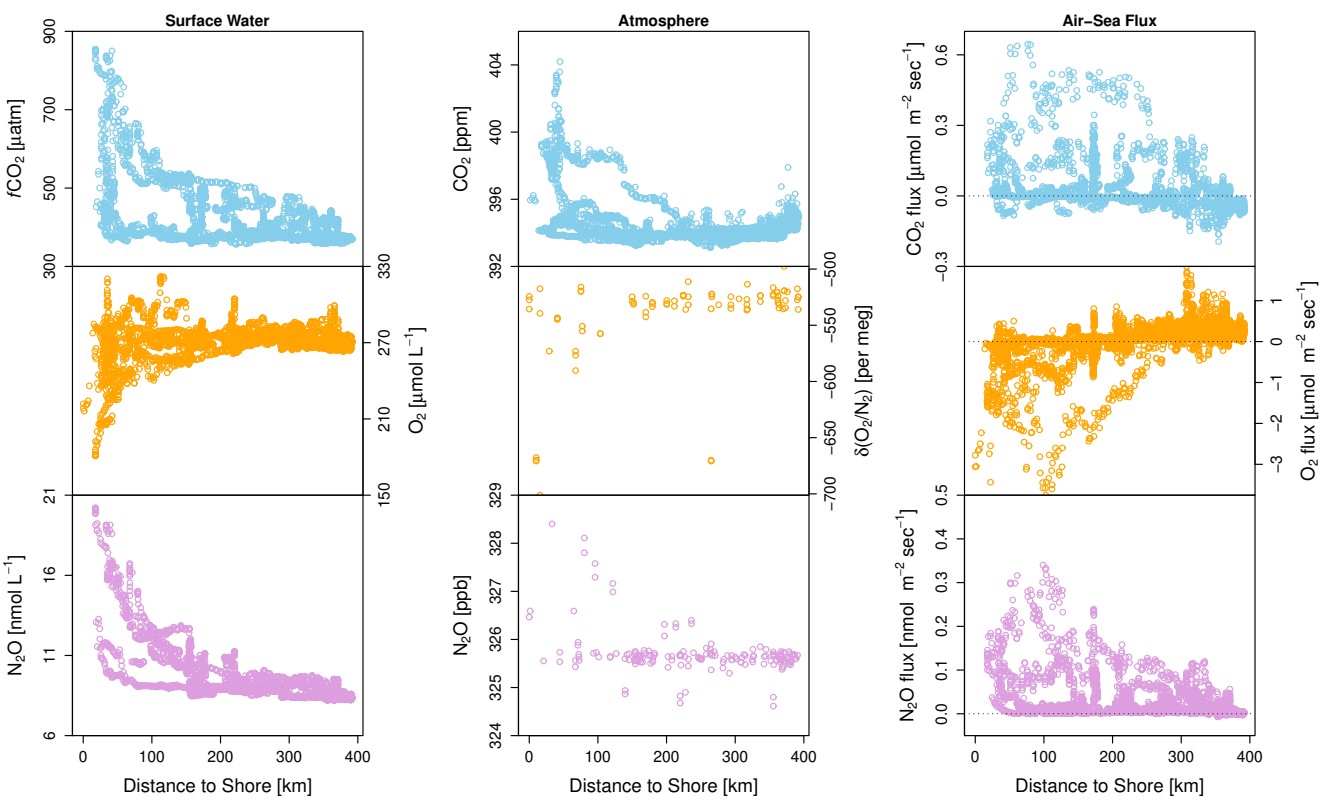

**Figure 7.** Dissolved concentrations of $CO_2$, $O_2$, and $N_2O$ in surface water, ca. 6 m depth during M99 (*left panel*), the atmospheric abundance of all three species (*middle panel*), and air–sea flux densities for $CO_2$, $O_2$, and $N_2O$ with $k_{W14}$ (*right panel*), all as a function of distance from shore. The dashed line on the rightmost panel indicates equilibrium between seawater and the atmosphere (i.e. no flux).

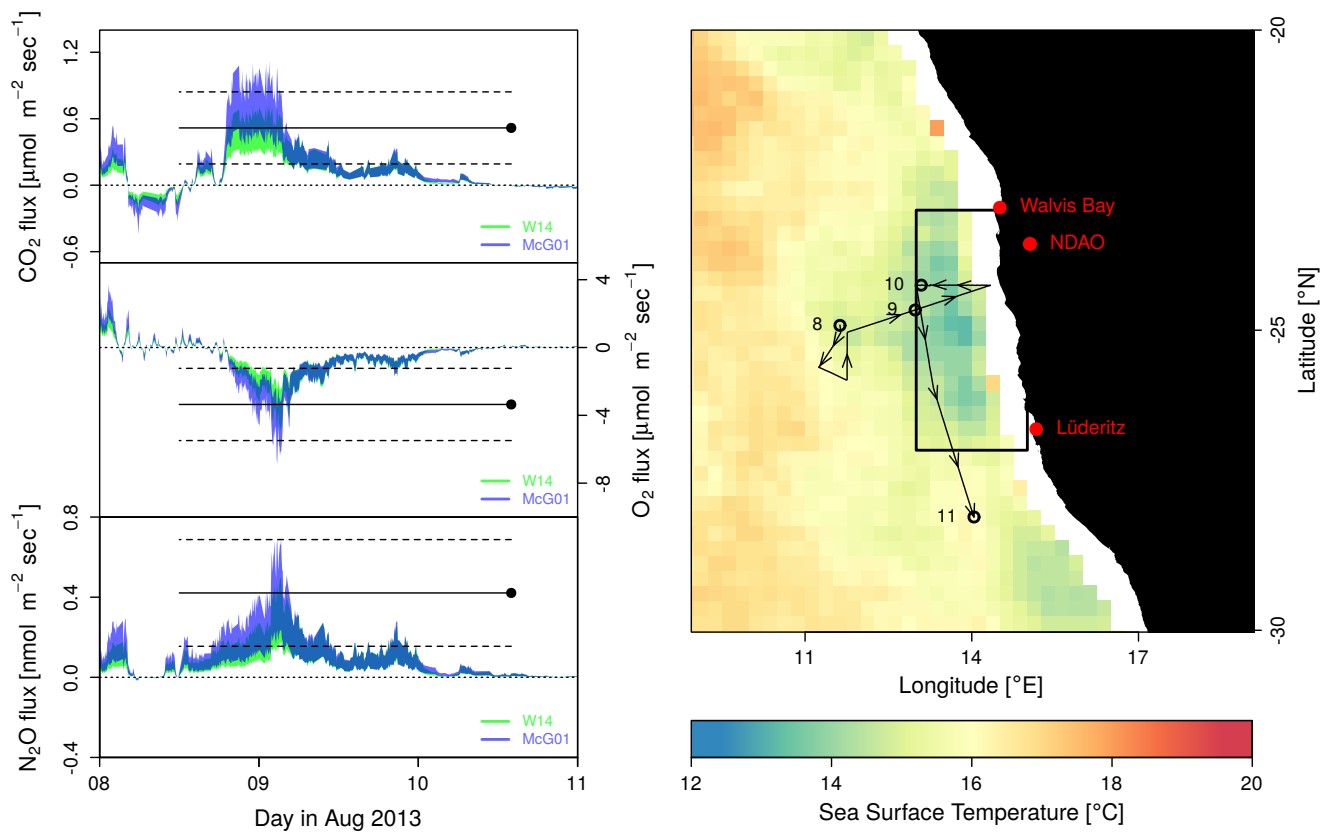

**Figure 8.** Air–sea flux densities for $CO_2$, $O_2$, and $N_2O$ using bottom-up methods (*left panel*), with a shaded envelope depicting the estimated surface flux and its uncertainty. Three estimates are shown, each made with a different parameterization for $k_w$. A positive value indicates net evasion. W14 and McG01 refer to the specific gas transfer velocity parameterization used. The top-down flux density estimate is plotted as a dot at the time of peak of the associated atmospheric anomaly. The horizontal line extending from each dot represents the time period during which the flux density associated with the anomaly was estimated to have occurred. Dotted lines indicate the uncertainty of the top-down estimate. Grid-cell average TRMM SST data for the three-day period is overlain with a cruise track and the Lüderitz/Walvis Bay domain (*right panel*). The days in August 2013 are marked with labels and open circles on the cruise track.

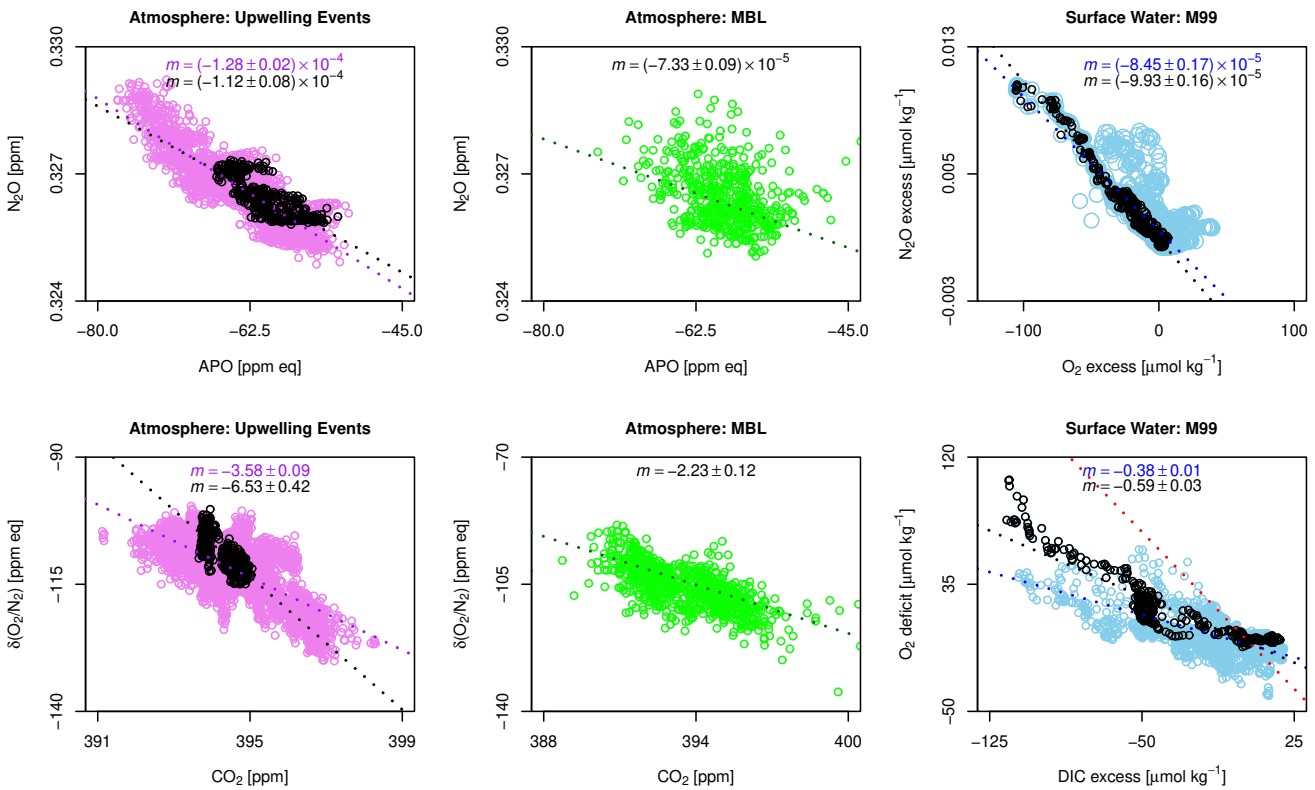

**Figure 9.** Comparison of the variability of $O_2$ with respect to $N_2O$ and $CO_2$ with respect to $O_2$ at NDAO and in surface water. Displayed are the data corresponding to atmospheric anomalies associated with upwelling events (*left*), of all marine boundary layer air masses as selected by back-trajectories (*center*), and dissolved concentrations of $CO_2$, $N_2O$, and $O_2$ during M99 (*right*). Atmospheric $O_2$ is expressed as APO in ppm equivalents, and dissolved concentrations are expressed as the difference between the measured concentration and the concentration at saturation, *i.e.*, an excess, except for the bottom right panel, where it is shown as a deficit. In that plot, the Redfield ratio of 1.45 is plotted as a dotted red line for reference. Slopes (*m*) are given at the top of each plot. The black circles correspond with the upwelling event encountered during M99 (see Figure 8).