# Peer review of "Air—Sea Fluxes of Greenhouse Gases and Oxygen in the Northern Benguela Current Region During Upwelling Events"

_Biogeosciences, 2019_

## Referee Comment (RC1) · Anonymous Referee #1 · 26 May 2019

Morgan et al. present an evaluation of using ground-based measurements of gas concentrations to estimate air-sea gas fluxes in coastal waters off the northern Benguela upwelling region. This approach can provide greater resolution than ship-based sampling during upwelling events, and thus represents a valuable tool for monitoring air-sea gas flux in coastal upwelling areas, which are important sites for emitting greenhouse gases such as N2O and CH4 to the atmosphere, but may be subject to seasonal and short term variability that may be missed by infrequent ship-based sampling efforts. The authors validate their method by comparing ground-based flux estimates to those estimated from ship-based sampling, which is an important step. The manuscript goes on to discuss the sources of variability and error in their data, and ways in which data

must be filtered to minimize error. The authors ultimately suggest that tower-based flux montoring systems should be widely adopted to measure air-sea flux in eastern boundary upwelling systems. This work thus represents an important contribution to coastal marine air-sea flux studies, and should be published with minor revisions. The presentation quality and writing is excellent, and I have only a few minor comments/questions that need to be addressed (see below). While I agree that this presents an important contribution and should be widely adopted, I can't help but struggle with the fact that the towers only appear to be used to estimate fluxes during upwelling events, and may be being under-utilized. Could the authors briefly discuss other potential benefits of these towers? E.g. could they be used to measure downward fluxes as well? Or perhaps measuring land-air fluxes? These towers might also naturally be compatible with eddy-covariance flux estimates.

Specific Comments:

Page 2, Line 32 – can you give the approximate frequency at which measurements were taken? What standards were used, and how frequently the instruments were calibrated?

Page 3, line 4 – perhaps include a brief explanation for presenting atmospheric oxygen relative to N2? What is this correcting for?

Page 3, line 7 – remove one of the 'as' (there are two in a row)

Page 3 line 16 – Why not include areas further off-shelf? Perhaps a justification for selecting these boundaries would be helpful? i.e. it's upstream of your tower in the direction of prevailing winds during upwelling? Or perhaps it's based on SST anomalies during upwelling events?

Page 3, line 19 – how were wind-speeds and SSTs determined? Update: I see this is discussed in the subsequent section. It's nice to understand where the data come from before you explain how the data was used, so I suggest moving the discussion on

[Figure]

Remote Sensing Data ahead of this section. But this is only a suggestion.

Page 3 – line 19-25 – I understand that you chose the upwelling thresholds by visually inspecting the data, but perhaps you could clarify how 'extreme' the thresholds you chose were relative to the standard deviation from your smoothed SST or wind-speed curves? Or perhaps you could state how many upwelling events were flagged during this step before filtering them further based on the SST, atmospheric data, back-trajectories of winds, and CO concentration?

Page 5 section 2.4. I'm curious about the dilution factor calculation – q. This seems straightforward, but I'm sure there are complications that may be being glossed over, and I have a few questions that I feel should be addressed in the text. How many sets of data points from the ship and tower were used to derive the average q? What is meant by 'filtered to exclude for poor agreement between $CO_2$ and $CH_4$'? and what percentage of the potentially viable measurements were excluded from the average because of this? And finally, how does the q you derived compare with other estimates?

Page 8 line 32 to end of para – you suggest that upwelling fluxes probably account for the majority of the mean annual flux (as reported by others). Can you support this claim by calculating how many upwelling events similar to the ones you observed (i.e. flux density x duration of UW event) would be needed to account for the annual flux assuming the fluxes the rest of the year were net-zero?

Page 9 line 3-6 – Good! I was wondering about how different $O_2/N_2$ vs. APO would be. Thank you for including this... , but where are the actual comparison between $dO_2/N_2$ and APO shown? There isn't a reference to a figure or table here, nor a slope and $r^2$ value, or even a mean difference between the two.

Page 9 line 10 – can you include the range here as well? It would be useful to know the maximum flux density you measured to compare with the others' estimates mentioned later in the paragraph. Figure 1 – can you please show the locations of Walvis Bay and Luderitz on the maps, since you refer to them in the text?

Page 9 line 25 – what were the dissolved CH4 concentrations in the upper 15m during this cruise? How do they compare with the other concentration values reported in the same paragraph?

Page 10, line 1 and 2. This sentence seems to contradict the second sentence of the paragraph, which states you observed elevated CH4 mixing ratios. Something is unclear here.

Page 10, line 18 – 'likely a result of warming temperatures that would reduce their solubility' are you suggesting the only reason the gases evaded to the atmosphere was warming? This implies CH4 and CO2 weren't supersaturated before the water warmed up...is that true? More likely the net evasion was enhanced by warming, except for O2.

Conclusion – Might be worth emphasizing the fact that your tower method is capable of measuring methane flux associated with both bubble flux and diffusion, which can't be said of ship-based measurements.

As previously mentioned, I agreed that a network of these towers would be great, but it might be easier to sell the idea of a network of these towers by mentioning the other potential uses of these towers. E.g. can you not estimate air-sea fluxes during down-welling events? What about land-air fluxes when winds from the east? You already mentioned the filtering for biomass burning based on CO, so surely there is some value here as well?

---

## Referee Comment (RC2) · Anonymous Referee #2 · 28 May 2019

The manuscript by Morgan et al., discusses a new dataset of land-based observations of greenhouse gases (CO2, CH4, N2O) and oxygen (O2) from the Benguela upwelling region. These observations are used to estimate air-sea fluxes of these gases during upwelling events, which are then compared to traditional ship-based estimates, suggesting an overall agreement between the different methods. The flux reconstructions are also used to discuss the processes underlying the cycles of these gases in the upwelling system, and to argue that the region hosts a significant source of CH4, presumably related to sedimentary production.

The main finding of the paper is that during periods of upwelling, identified from

satellite-based SST observations, land-based measurements of greenhouse gases show positive excursions of CO2, CH4, N2O (and possibly CO), and negative excursions in O2. These atmospheric concentrations are then translated with a simple transport model into air-sea fluxes of these gases, resulting into fluxes from the ocean to the atmosphere for greenhouse gases, and from the atmosphere to the ocean for O2. This picture is broadly consistent with our understanding of upwelling systems, and is supported by analysis of the stoichiometry of these fluxes, e.g. N2O:O2. The air-sea fluxes estimated by this top-down method are further compared with ship-based estimates for a particular upwelling event during which in-situ observations were collected, suggesting consistency between the (very different) approaches. This broad consistency is used to advocate for continues monitoring of greenhouse gas air-sea fluxes by land-based stations in this and other upwelling systems.

The Authors present a novel dataset of atmospheric measurements and a fairly through analysis that connect them to air-sea fluxes from the region. The focus of the paper are upwelling events, when the signature of air-sea gas exchange is particularly recognizable in the nearby atmosphere. Overall, the methods and type of data discussed are not completely new, but the application to the Benguela upwelling system is, and the Authors present an argument for the usefulness of this type of measurements. Upwelling systems are regions of enhanced exchange between the ocean interior and the atmosphere, and important components of global greenhouse gas budgets (e.g. N2O and CH4). Thus, the study connects to a topic of global relevance that should be of interest to a broader readership in oceanic/atmospheric biogeochemistry. The paper is overall well written, and the figures informative. The interpretation of the data is fairly clear and overall sound, and the paper in principle suitable for publication in Biogeosciences. That said, I have a several comments that I think should be addressed before the paper is ready for publication.

The model used to estimate the air-sea flux densities (Section 2.4 and equation 3) is essential for the top-down estimates presented in the paper. While the model is

fairly simple, it is not particularly well described and critically assessed (see specific comment below). In particular, no uncertainty estimate arising from the model itself is discussed or quantified. The only air-sea flux estimate uncertainties appear to be standard deviations from all estimates, but the model presumably introduces inherent uncertainty in each estimate, which should be quantified. In particular, assumptions on the mixing constant "q" may be particularly impactful.

The Authors claim that the resulting top-down fluxes agree well with the in situ, ship-based estimates, even though in practice there's only one event for which the comparison is possible. However, the top-down estimates seem to be systematically higher than the ship-based estimates (e.g. in Fig. 8). While indicating "order-of magnitude" agreement at the peak outgassing (ingassing for O2), the figure points to a potential overestimate of the top-down method for all gases, especially if one integrates the fluxes over time (e.g. over the period indicated with the horizontal bar in Fig. 8). This is somewhat surprising, because the top-down method should integrate over broader regions with compensations between high and low fluxes. This discrepancy should be reconciled or more thoroughly discussed, but it feels minimized by the Authors (e.g. "good agreement" in the abstract and in few parts of the main text), which is a disservice to the readers. The Authors could be more nuanced with the discussion of this comparison, and more forward with the limitations of the approach, and discuss ways to address them, if the objective is a credible extension and application in future studies e.g. for monitoring and quantitative estimates.

Related to the point above, while the Authors are clear that the comparison is not a calibration of the top-down method, at some point such calibration will be needed to make the estimates quantitative and reliable, thus it would be useful if the Authors could add a discussion of the possible work and steps needed to turn this comparison exercise into a credible approach that could be used for monitoring. This type of discussion would strengthen the conclusions (page 13, lines 3-6), which right now feel somewhat superficial.

The rationale for focusing the study on upwelling events could be better explained earlier in the paper. I assume it is related to the strength of the signal to be detected, stronger during upwelling, which allows a first demonstration of the method, but I may be wrong.

In the same way, the study falls a bit short of fully connecting regional results to the big picture of greenhouse air-sea fluxes in upwelling systems. While it is interesting and valuable to provide air-sea flux estimates for all upwelling events in the region (Table 1), it would be even more relevant to couch these estimates into the big picture of air-sea fluxes for the region. For example, is outgassing of greenhouse gases during upwelling region important for total gas budgets? Could upwelling events be responsible for most of the outgassing, or is outgassing during non-upwelling periods also important? Of course, this would require comparison with other large-scale estimates for the region, and some degree of extrapolation/speculation, but it could add breath to the paper.

Page 2, line 9: why "yearly"? It seems that this approach could be applied to any timescale long enough to encompass the observations utilized. Please clarify or remove.

Page 3, line 6 and following: I see the point of utilizing APO, however its introduction is somewhat abrupt and not every reader may be familiar with the concept and scope of it. I suggest a sentence or two to clarify and explain the usefulness of this tracer in the context of the paper (it is only affected by air-sea gas flux differences between O2 and CO2). A justification is presented later (e.g. page 9) but it could be more useful early on.

Page 3, lines 19-25. I wonder if any consideration was given to including the direction of wind in the upwelling detection algorithm, since Ekman theory implies that only favorable wind directions (here equatorward and parallel to the coast) would induce upwelling. This could be clarified.

Section 2.4. The model rationale, variables, uncertainty, and limitation should be discussed in better detail, as it form the basis for the top-down air-sea flux estimates. First of all, it is unclear what is solved for (I assume F) and how, e.g. based on what other quantities. Second, it is not clear how the back-trajectories were determined (there is no discussion of it that I could find) and how they are used in the model — I presume the variable "x" is the distance along these trajectories. The atmospheric boundary layer is assumed to be constant with a thickness "h", but this possibly varies substantially on a variety of temporal scales, e.g. going from he ocean to the land, and over the course of a day. Maybe variations in h can be folded into variations in the mixing rate q, but this rate is assumed to be constant, which is a big assumption. More critically, q is determined from equation (4), which presumably is a derivation of equation (3), although my sense is that it can be only derived if one assumes F=0 for the two gases, which is inaccurate. In equation (4) it is not clear what "t" represents and how it was determined (I suppose from x/U). The determination of "q" seems critical for the method, and it should be discussed in more detail, and results shown, for example of the determination of q for CO2, CH4, etc. Uncertainty in q could then be propagated into the model, or at least its effect on the flux estimates discussed.

Page 6, line 12. The method by Lee et al. is somewhat outdated, and has been superseded by more recent approaches, e.g. the "LIAR" method, Carter et al., 2016, Limn. Ocean. Methods, although I suspect the alkalinity approximation is not a major source of error in the CO2 flux calculation.

Page 7, lines 6-15. The choice of piston velocity formulation seems hazardous, and needs some justification and perhaps clarifications. The Wanninkhof 1992 formulation has been superseded by a more recent one in Wanninkhof 2014, Limn. Ocean. Methods. The old formulation is biased too high by approximately 20% and should not be used. The Nightingale 2000 formulation is an odd choice because it was developed for the North Sea, and its range of validity is 3 to 14 m/s (wind speeds used in this paper can be smaller than that). The paper by Roobaert et al., 2017, Biogeosciences, provides useful guidelines for the choice of piston velocity that could be considered in

the study.

Page 7, line 13, "kw and U10 must be in the same units, e.g. m sec-1": this is incorrect. With the coefficient reported, U10 must be expressed in m/s, and kw in cm/hour. Effectively, the coefficients have units, e.g. cm/h/(m2/s2) for the quadratic coefficient, etc.

Page 8, lines 3-4: clarify the difference between "identifying" and "detecting" an upwelling event, otherwise the sentence is not clear.

Page 8, line 14: this sentence begins discussing an upwelling event, but then two are mentioned. Please clarify the time for the second event (presumably Dec. 4th)

Page 9, line 24: please provide a reference or some context for the GENUS cruise.

Page 9, line 32: is this "synoptic event" also corresponding to an upwelling event?

Page 9, line 34: "This coincides with . . .". Please clarify this sentence; it is not clear what "this" refers to.

Page 13, line 3-6: "Based on our results . . .". Measuring programs of this type are already in place in few regions, e.g. as part of the Advanced Global Atmospheric Gases Experiment (AGAGE). For example there are two land-based monitoring stations in the California upwelling region. It may be useful to expand this part of the conclusions to acknowledge existing observational programs and previous work, discussing what has been learned from them, and what is still missing (e.g. spatial coverage over other upwelling systems?). It would also be useful if the Authors could speculate on how far this type of measurements can go in order to provide truly quantitative estimates of air sea fluxes from coastal upwelling regions, since the paper only provides a proof of concept that still suffers from very large uncertainties.

Page 5, line 26: "deviated" from what?

Figure 1, and Fig. 8: it would be useful to add to the maps a few geographic reference

points, e.g. the town of Luderitz, which is mentioned multiple times in the manuscript, for the readers who are not familiar with the region.

Figure 1, left panels, and Fig. 4: please highlight the upwelling events as detected by the algorithm used, e.g. with vertical bars or shadings.

Figure 3: please highlight, the periods corresponding to the M99 cruise, and the upwelling events shown in Fig. 1, e.g. either at the top/bottom of the figure, or using bar shadings of a different color. Why is chl-a shown as dots instead of as a continuous line?

Figure 5: please clarify the duration of the back trajectory periods.

Figure 8, right panel: it would be useful to mark the days on the cruise track, to allow a comparison with the left panels.

Figure 9, left panels: the gray symbols are very hard to see on the purple background, please use a different color (e.g. darker). At the end of the caption: "correspond to" instead of "correspond with".

Please clarify early in the paper what "bottom-up" and "top-down" estimates refer to. This terminology in the specific context of the paper may not be clear to every reader. E.g. in Fig. 8, the Authors could add a clarification on "bottom-up" (ship-based) and "top-down" (land-based).

---

## Author Comment (AC1) · 11 Jun 2019

Morgan et al. present an evaluation of using ground-based measurements of gas concentrations to estimate air-sea gas fluxes in coastal waters off the northern Benguela upwelling region. This approach can provide greater resolution than ship-based sampling during upwelling events, and thus represents a valuable tool for monitoring air-sea gas flux in coastal upwelling areas, which are important sites for emitting greenhouse gases such as N2O and CH4 to the atmosphere, but may be subject to seasonal and short term variability that may be missed by infrequent ship-based sampling efforts.

The authors validate their method by comparing ground-based flux estimates to those estimated from ship-based sampling, which is an important step. The manuscript goes on to discuss the sources of variability and error in their data, and ways in which data must be filtered to minimize error. The authors ultimately suggest that tower-based flux montoring systems should be widely adopted to measure air-sea flux in eastern boundary upwelling systems. This work thus represents an important contribution to coastal marine air-sea flux studies, and should be published with minor revisions. The presentation quality and writing is excellent, and I have only a few minor comments/questions that need to be addressed (see below).

– We would like to thank the referee for their time and constructive review of our manuscript. We include our responses to their comments below.

While I agree that this presents an important contribution and should be widely adopted, I can't help but struggle with the fact that the towers only appear to be used to estimate fluxes during upwelling events, and may be being under-utilized. Could the authors briefly discuss other potential benefits of these towers? E.g. could they be used to measure downward fluxes as well? Or perhaps measuring land-air fluxes? These towers might also naturally be compatible with eddy-covariance flux estimates.

– The reviewer brings up an important point to clarify, which we will address in the Introduction and Conclusion. We focus on upwelling events because they are detectable in the atmospheric record due to their distinct tracer-tracer relationships. The tower data can be used to quantify regional, time-varying surface fluxes for both land and ocean, but this would ultimately require a Bayesian atmospheric inversion, which is beyond the scope of our study. We will add this text to the Introduction: "We focus on individual upwelling events as we expect them to be distinguishable from other sources of intraseasonal variability based their apparent stoichiometry in the atmosphere, and because there are relatively few observation-based studies from this region, relative to other EBUS."

Specific Comments:

Page 2, Line 32 – can you give the approximate frequency at which measurements were taken? What standards were used, and how frequently the instruments were calibrated?

– Such details were omitted for brevity, as they were published elsewhere, but we will add the requested information.

Page 3, line 4 – perhaps include a brief explanation for presenting atmospheric oxygen relative to N2? What is this correcting for?

– We will add the clarification that we are following a standard convention here. There are several reasons to report the $O_2/N_2$ ratio instead of the absolute mole fraction, one being that $O_2$ is not a trace gas, and the mole fraction of $O_2$ will vary due to the addition or subtraction of other trace gases in a given parcel of air to a non-negligible degree. A common thought experiment demonstrating this: in a parcel of air containing 1e6 molecules of air, 410 of which are $CO_2$ and 209,392 of which are $O_2$, the mole fractions will be 410 ppm and 0.209392, respectively. Adding one molecule of $CO_2$ to the parcel changes the mole fraction by 0.999589 ppm for $CO_2$ but 0.2093918 ppm for $O_2$.

Page 3, line 7 – remove one of the 'as' (there are two in a row)

– We will remove.

Page 3 line 16 – Why not include areas further off-shelf? Perhaps a justification for selecting these boundaries would be helpful? i.e. it's upstream of your tower in the direction of prevailing winds during upwelling? Or perhaps it's based on SST anomalies during upwelling events?

– We will add the following text: "We selected this domain because it represented an area of the coast where strong upwelling occurs regularly (Demarcq et al, 2007), where this upwelling was spatially distinct from other upwelling cells reported in the literature (Lutjeharms and Meeuwis, 1987; Veitch et al, 2009), and where upwelling

was downwind of the station during upwelling events. These criteria were considered desirable because they would provide the best opportunities for relating atmospheric anomalies to upwelling events. These determinations were based on analysis of our SST dataset, and atmospheric back-trajectories simulated with the HYSPLIT model." In addition, we have added a brief description of the HYSPLIT model, for completeness.

Page 3, line 19 – how were wind-speeds and SSTs determined? Update: I see this is discussed in the subsequent section. It's nice to understand where the data come from before you explain how the data was used, so I suggest moving the discussion on Remote Sensing Data ahead of this section. But this is only a suggestion.

– We will move the section up.

Page 3 – line 19-25 – I understand that you chose the upwelling thresholds by visually inspecting the data, but perhaps you could clarify how 'extreme' the thresholds you chose were relative to the standard deviation from your smoothed SST or wind-speed curves? Or perhaps you could state how many upwelling events were flagged during this step before filtering them further based on the SST, atmospheric data, back-trajectories of winds, and CO concentration?

– We will add standard deviations of both anomaly time series to the Methods section. We will also add additional details about the number of filtered/excluded of events, and move all of that text to the Results section, since this seems a more appropriate place for it.

Page 5 section 2.4. I'm curious about the dilution factor calculation – q. This seems straightforward, but I'm sure there are complications that may be being glossed over, and I have a few questions that I feel should be addressed in the text. How many sets of data points from the ship and tower were used to derive the average q? What is meant by 'filtered to exclude for poor agreement between CO2 and CH4'? and what percentage of the potentially viable measurements were excluded from the average because of this? And finally, how does the q you derived compare with other estimates?

– We agree this needs more discussion and clarification. The dilution factor q is probably the least constrained and well-known of all the model parameters. Since there were high-resolution atmospheric measurements of $CO_2$ and CH4 on the Meteor cruise, either species could be used to estimate a non-species dependent dilution factor. If the difference of $q^{CO_2}$ and $q^{CH_4}$ for a given comparison of was > ± 0.01, they were excluded from the average. This reduced the total n from 32 to 13. Another way to estimate q would be to look at particle dispersion rates in a Lagrangian Particle Dispersion Model (LPDM), which we can include in a revised manuscript. Either way, we will increase the uncertainty bounds on this parameter. We will also add some discussion of other published estimates, e.g. Price et al 2004 (JGR Atmos., 109, 23), who used multiple species and a model approach to arrive at a mean q of 0.010 ± 0.004 hr$^{-1}$, which is nearly identical to our value of 0.011 ± 0.006 hr$^{-1}$, although the spatial and temporal scale they considered was larger. Dillon et al 2002 (JGR Atmos., 107, D5) observed rates that were higher for a Sacramento pollution plume, 0.2 hr$^{-1}$. We will also broaden the discussion to include using q as a tuning parameter for the model – i.e., what value of q would best fit the shipboard flux estimates if all other model parameters were equal, and then compare this to our experimentally determined value of q, which is a conceptually different approach to the puff model. This would mean that the comparison to the shipboard estimate is approached on a different basis, i.e. as a means for improving the model rather than validating it.

Page 8 line 32 to end of para – you suggest that upwelling fluxes probably account for the majority of the mean annual flux (as reported by others). Can you support this claim by calculating how many upwelling events similar to the ones you observed (i.e. flux density x duration of UW event) would be needed to account for the annual flux assuming the fluxes the rest of the year were net-zero?

– We can provide a rough estimate of the total annual flux due to upwelling events for each species, by making some broad assumptions on the area of upwelled water, and then compare these flux totals, but we would like to restrict this to the Discussion, as

the results are not that robust, due to the simplifying assumptions.

Page 9 line 3-6 – Good! I was wondering about how different O2/N2 vs. APO would be. Thank you for including this. . . , but where are the actual comparison between dO2/N2 and APO shown? There isn't a reference to a figure or table here, nor a slope and r2 value, or even a mean difference between the two.

– We hesitate to add another figure (to keep the total number from becoming overly high), since it only shows a strong linear correlation between the two flux estimates, but we can add some text/statistics on the comparison: "A linear regression between the two estimated flux densities yielded a slope of 0.91 and a coefficient of determination of $R^2$ = 0.98. The estimated APO flux density was 4.5% lower on average than the $\delta(O_2/N_2)$-inferred flux density."

Page 9 line 10 – can you include the range here as well? It would be useful to know the maximum flux density you measured to compare with the others' estimates mentioned later in the paragraph.

– This will be added to the revised manuscript.

Figure 1 – can you please show the locations of Walvis Bay and Luderitz on the maps, since you refer to them in the text?

– Yes, this will be added to the revised manuscript.

Page 9 line 25 – what were the dissolved CH4 concentrations in the upper 15m during this cruise? How do they compare with the other concentration values reported in the same paragraph?

– We will add the text, "from dissolved concentrations ranging from 6.0 to 140 nM." FYI we have identified 6 other samples from the MEMENTO database, bringing the total number to 9, though the range reported hasn't changed.

Page 10, line 1 and 2. This sentence seems to contradict the second sentence of
the paragraph, which states you observed elevated CH4 mixing ratios. Something is unclear here.

– Thank you for pointing this out, we will clarify by adding "apart from the initial synoptic event".

Page 10, line 18 – 'likely a result of warming temperatures that would reduce their solubility' are you suggesting the only reason the gases evaded to the atmosphere was warming? This implies CH4 and CO2 weren't supersaturated before the water warmed up. . .is that true? More likely the net evasion was enhanced by warming, except for O2.

– We are not saying that the only thing influence the flux of the gases was warming, but that generally we would assume that a positive flux ratio between $O_2$ and $CO_2$ is a result of thermal processes dominating the surface flux, since net respiration/ventilation of aged waters or net productivity would produce a negative flux ratio. Given that it was strong enough to reverse the flux of $O_2$ relative to the rest of the filament, this interpretation seems to us to be justified. We will add the text, "The positive flux ratio of $O_2$:$CO_2$ is generally only produced when thermal processes dominate the air-sea flux."

Conclusion – Might be worth emphasizing the fact that your tower method is capable of measuring methane flux associated with both bubble flux and diffusion, which can't be said of ship-based measurements.

– This is an excellent point, although we are hesitant to include such a statement, since the uncertainties associated with our method are high.

As previously mentioned, I agreed that a network of these towers would be great, but it might be easier to sell the idea of a network of these towers by mentioning the other potential uses of these towers. E.g. can you not estimate air-sea fluxes during down-welling events? What about land-air fluxes when winds from the east? You already mentioned the filtering for biomass burning based on CO, so surely there is some

value here as well?

– Thanks for pointing this out, we will adjust the concluding text, similar to our first response above, by adding the following sentences: "We have focused here on upwelling events, because they are distinguishable from other sources of intraseasonal variability in the atmospheric record. A full top-down accounting of the greenhouse gas budget of the the Benguela could be accomplished through a Bayesian atmospheric inversion of one or more coastal stations."

---

## Author Comment (AC2) · 11 Jun 2019

The manuscript by Morgan et al., discusses a new dataset of land-based observations of greenhouse gases (CO2, CH4, N2O) and oxygen (O2) from the Benguela upwelling region. These observations are used to estimate air-sea fluxes of these gases during upwelling events, which are then compared to traditional ship-based estimates, suggesting an overall agreement between the different methods. The flux reconstructions are also used to discuss the processes underlying the cycles of these gases in the upwelling system, and to argue that the region hosts a significant source of CH4, pre-

sumably related to sedimentary production.

The main finding of the paper is that during periods of upwelling, identified from satellite-based SST observations, land-based measurements of greenhouse gases show positive excursions of CO2, CH4, N2O (and possibly CO), and negative excursions in O2. These atmospheric concentrations are then translated with a simple transport model into air-sea fluxes of these gases, resulting into fluxes from the ocean to the atmosphere for greenhouse gases, and from the atmosphere to the ocean for O2. This picture is broadly consistent with our understanding of upwelling systems, and is supported by analysis of the stoichiometry of these fluxes, e.g. N2O:O2. The air-sea fluxes estimated by this top-down method are further compared with ship-based estimates for a particular upwelling event during which in-situ observations were collected, suggesting consistency between the (very different) approaches. This broad consistency is used to advocate for continues monitoring of greenhouse gas air-sea fluxes by land-based stations in this and other upwelling systems.

The Authors present a novel dataset of atmospheric measurements and a fairly through analysis that connect them to air-sea fluxes from the region. The focus of the paper are upwelling events, when the signature of air-sea gas exchange is particularly recognizable in the nearby atmosphere. Overall, the methods and type of data discussed are not completely new, but the application to the Benguela upwelling system is, and the Authors present an argument for the usefulness of this type of measurements. Upwelling systems are regions of enhanced exchange between the ocean interior and the atmosphere, and important components of global greenhouse gas budgets (e.g. N2O and CH4). Thus, the study connects to a topic of global relevance that should be of interest to a broader readership in oceanic/atmospheric biogeochemistry. The paper is overall well written, and the figures informative. The interpretation of the data is fairly clear and overall sound, and the paper in principle suitable for publication in Biogeosciences. That said, I have a several comments that I think should be addressed before the paper is ready for publication.

– We thank the reviewer for their thorough and thoughtful review of our manuscript.

The model used to estimate the air-sea flux densities (Section 2.4 and equation 3) is essential for the top-down estimates presented in the paper. While the model is fairly simple, it is not particularly well described and critically assessed (see specific comment below). In particular, no uncertainty estimate arising from the model itself is discussed or quantified. The only air-sea flux estimate uncertainties appear to be standard deviations from all estimates, but the model presumably introduces inherent uncertainty in each estimate, which should be quantified. In particular, assumptions on the mixing constant "q" may be particularly impactful.

– We agree with the reviewer's assessment. For the original paper, uncertainties for each term were calculated or estimated and propagated in quadrature sums; this is the source of the dotted lines on the top-down estimate in Figure 8. We will revisit these uncertainties and present them in a table with relevant statistics.

The Authors claim that the resulting top-down fluxes agree well with the in situ, ship-based estimates, even though in practice there's only one event for which the comparison is possible. However, the top-down estimates seem to be systematically higher than the ship-based estimates (e.g. in Fig. 8). While indicating "order-of magnitude" agreement at the peak outgassing (ingassing for O2), the figure points to a potential overestimate of the top-down method for all gases, especially if one integrates the fluxes over time (e.g. over the period indicated with the horizontal bar in Fig. 8). This is somewhat surprising, because the top-down method should integrate over broader regions with compensations between high and low fluxes. This discrepancy should be reconciled or more thoroughly discussed, but it feels minimized by the Authors (e.g. "good agreement" in the abstract and in few parts of the main text), which is a disservice to the readers. The Authors could be more nuanced with the discussion of this comparison, and more forward with the limitations of the approach, and discuss ways to address them, if the objective is a credible extension and application in future studies e.g. for monitoring and quantitative estimates.

– This is a good point; the method tends to overestimate fluxes as compared to the shipboard measurements, and because we can only identify upwelling evens with anomalies that are above the baseline, smaller events are not included in our average flux densities. We will broaden our discussion of the limitations of our approach.

Related to the point above, while the Authors are clear that the comparison is not a calibration of the top-down method, at some point such calibration will be needed to make the estimates quantitative and reliable, thus it would be useful if the Authors could add a discussion of the possible work and steps needed to turn this comparison exercise into a credible approach that could be used for monitoring. This type of discussion would strengthen the conclusions (page 13, lines 3-6), which right now feel somewhat superficial.

– This is a good suggestion. We will add some text that discusses necessary steps for continuous top-down monitoring.

The rationale for focusing the study on upwelling events could be better explained earlier in the paper. I assume it is related to the strength of the signal to be detected, stronger during upwelling, which allows a first demonstration of the method, but I may be wrong.

– The motivation was that we were interested in upwelling as a potential source of GHGs, and their episodic nature and distinctive signatures made it possible to estimate the flux without performing a full inversion of 3+ species from a single station. We have added this sentence to the introduction: "We focus on individual upwelling events as we expect them to be distinguishable from other sources of intraseasonal variability based their apparent stoichiometry in the atmosphere, and because there are relatively few observation-based studies from this region, relative to other EBUS."

In the same way, the study falls a bit short of fully connecting regional results to the big picture of greenhouse air-sea fluxes in upwelling systems. While it is interesting and valuable to provide air-sea flux estimates for all upwelling events in the region (Table 1),

it would be even more relevant to couch these estimates into the big picture of air-sea fluxes for the region. For example, is outgassing of greenhouse gases during upwelling region important for total gas budgets? Could upwelling events be responsible for most of the outgassing, or is outgassing during non-upwelling periods also important? Of course, this would require comparison with other large-scale estimates for the region, and some degree of extrapolation/speculation, but it could add breath to the paper.

– We can provide rough estimates of the annual flux of each species from the study region, but this will require assuming a constant flux density and knowing the total area of upwelled water, simplifying assumptions which we can bracket with appropriate uncertainties, but will not be terribly robust. So, we will add this to the Discussion section, rather than present them in Results.

Page 2, line 9: why "yearly"? It seems that this approach could be applied to any timescale long enough to encompass the observations utilized. Please clarify or remove.

– We will remove.

Page 3, line 6 and following: I see the point of utilizing APO, however its introduction is somewhat abrupt and not every reader may be familiar with the concept and scope of it. I suggest a sentence or two to clarify and explain the usefulness of this tracer in the context of the paper (it is only affected by air-sea gas flux differences between O2 and CO2). A justification is presented later (e.g. page 9) but it could be more useful early on.

– We will revise the text introducing APO to read as follows: "In order to isolate the influence of air-sea exchanges on $O_2/N_2$, we have employed the use of a data-derived tracer, known as atmospheric potential oxygen (APO), which masks variations of $O_2/N_2$ that are due to terrestrial biosphere exchange (Stephens et al, 1998). Variations in APO are thus primarily due to fossil fuel burning and air-sea gas exchange of $O_2$. APO is defined as:"

Page 3, lines 19-25. I wonder if any consideration was given to including the direction of wind in the upwelling detection algorithm, since Ekman theory implies that only favorable wind directions (here equatorward and parallel to the coast) would induce upwelling. This could be clarified.

– We did not explicitly consider wind direction in our detection algorithm, but consideration was given to wind direction at the station (selection for sea breeze) and air mass origin (back trajectory filtering). These two steps together effectively filtered out all events which were not upwelling-favorable. Only upwelling-favorable winds would bring air to the station from these two upwelling cells, which is part of the reason for their selection. This is specified in the text now: "We selected this domain because it represented an area of the coast where strong upwelling occurs regularly (Demarcq et al, 2007), where this upwelling was spatially distinct from other upwelling cells reported in the literature (Lutjeharms and Meeuwis, 1987; Veitch et al, 2009), and where upwelling was downwind of the station during upwelling events."

Section 2.4. The model rationale, variables, uncertainty, and limitation should be discussed in better detail, as it form the basis for the top-down air-sea flux estimates. First of all, it is unclear what is solved for (I assume F) and how, e.g. based on what other quantities. Second, it is not clear how the back-trajectories were determined (there is no discussion of it that I could find) and how they are used in the model â ËŸAËĞT I presume the variable "x" is the distance along these trajectories.

– We can add some text on the HYSPLIT model. That is correct, $x$ is the distance along the trajectory. We will clarify this in the text.

The atmospheric boundary layer is assumed to be constant with a thickness "h", but this possibly varies substantially on a variety of temporal scales, e.g. going from the ocean to the land, and over the course of a day. Maybe variations in h can be folded into variations in the mixing rate q, but this rate is assumed to be constant, which is a big assumption. More critically, q is determined from equation (4), which presumably

is a derivation of equation (3), although my sense is that it can be only derived if one assumes F=0 for the two gases, which is inaccurate. In equation (4) it is not clear what "t" represents and how it was determined (I suppose from x/U). The determination of "q" seems critical for the method, and it should be discussed in more detail, and results shown, for example of the determination of q for CO2, CH4, etc. Uncertainty in q could then be propagated into the model, or at least its effect on the flux estimates discussed.

– We will broaden this discussion in both the Methods and Results. One does need to assume F=0 for both gases, which is not accurate for the marine environment, but an acceptable assumption for the terrestrial component, as the terrain between the station and coast is devoid of vegetation or human settlement. In light of both reviewers' comments, we will compare our estimate of q to published values, and also conduct some simulations with a particle dispersion model to estimate q. We can then tune the q parameter to best match the shipboard measurements, and compare this with our estimates of q.

Page 6, line 12. The method by Lee et al. is somewhat outdated, and has been superseded by more recent approaches, e.g. the "LIAR" method, Carter et al., 2016, Limn. Ocean. Methods, although I suspect the alkalinity approximation is not a major source of error in the CO2 flux calculation.

– Thank you for bringing this to our attention. We can recalculate with the LIAR method.

Page 7, lines 6-15. The choice of piston velocity formulation seems hazardous, and needs some justification and perhaps clarifications. The Wanninkhof 1992 formulation has been superseded by a more recent one in Wanninkhof 2014, Limn. Ocean. Methods. The old formulation is biased too high by approximately 20% and should not be used. The Nightingale 2000 formulation is an odd choice because it was developed for the North Sea, and its range of validity is 3 to 14 m/s (wind speeds used in this paper can be smaller than that). The paper by Roobaert et al., 2017, Biogeosciences, provides useful guidelines for the choice of piston velocity that could be considered in

the study.

– We included the W92 because it is still widely in use, but are happy to replace it with W14. We thank the reviewer for bringing the Roobaert publication to our attention. We favored N00 because it was empirically derived in a coastal region, but we can include other $k_w$ parameterization(s) that better encompasses the wind speeds of our flux event (2 - 15 m sec$^{-1}$). For the direct comparison of peak observed shipboard flux, winds were in the ca. 12-15 m sec$^{-1}$ range.

Page 7, line 13, "kw and U10 must be in the same units, e.g. m sec-1": this is incorrect. With the coefficient reported, U10 must be expressed in m/s, and kw in cm/hour. Effectively, the coefficients have units, e.g. cm/h/(m2/s2) for the quadratic coefficient, etc.

– Thank you for bringing this to our attention, this was some text that got confused with $k_w$ being in m sec$^{-1}$ for Eq 7. We will change the text to read: "The Schmidt number is dimensionless, and U10 is in units of m sec$^{-1}$, producing $k_w$ in units of cm hr$^{-1}$."

Page 8, lines 3-4: clarify the difference between "identifying" and "detecting" an upwelling event, otherwise the sentence is not clear.

– We will change the text to: "detection of an anomaly in the atmospheric time series", the distinction being that we identify upwelling events based on SST, wind speed, etc, but can only detect them in the atmospheric time series when atmospheric transport allowed.

Page 8, line 14: this sentence begins discussing an upwelling event, but then two are mentioned. Please clarify the time for the second event (presumably Dec. 4th)

– Text now reads: "4th of December, when SST dropped again. During these two low SST pulses, chl-a values were higher."

Page 9, line 24: please provide a reference or some context for the GENUS cruise.

– Will do.

Page 9, line 32: is this "synoptic event" also corresponding to an upwelling event?

– No, the upwelling event began after the synoptic event. We will clarify this in the text.

Page 9, line 34: "This coincides with . . .". Please clarify this sentence; it is not clear what "this" refers to.

– We will clarify, "this" referred to the sporadic enhancements.

Page 13, line 3-6: "Based on our results . . .". Measuring programs of this type are already in place in few regions, e.g. as part of the Advanced Global Atmospheric Gases Experiment (AGAGE). For example there are two land-based monitoring stations in the California upwelling region. It may be useful to expand this part of the conclusions to acknowledge existing observational programs and previous work, discussing what has been learned from them, and what is still missing (e.g. spatial coverage over other upwelling systems?). It would also be useful if the Authors could speculate on how far this type of measurements can go in order to provide truly quantitative estimates of air sea fluxes from coastal upwelling regions, since the paper only provides a proof of concept that still suffers from very large uncertainties.

– We can add discussion of existing stations, and deepen our discussion of top-down quantification of coastal marine fluxes using atmospheric data.

Page 5, line 26: "deviated" from what?

– The cavity pressure setpoint of 140 torr, we will clarify.

Figure 1, and Fig. 8: it would be useful to add to the maps a few geographic reference points, e.g. the town of Luderitz, which is mentioned multiple times in the manuscript, for the readers who are not familiar with the region.

– Will do.

Figure 1, left panels, and Fig. 4: please highlight the upwelling events as detected by the algorithm used, e.g. with vertical bars or shadings.

– Will do.

Figure 3: please highlight, the periods corresponding to the M99 cruise, and the upwelling events shown in Fig. 1, e.g. either at the top/bottom of the figure, or using bar shadings of a different color. Why is chl-a shown as dots instead of as a continuous line?

– Will do. We can change the chl-a to a line, I think this plotting choice was made because there were more outliers than the other time series, and it made the plot harder to read with the shading.

Figure 5: please clarify the duration of the back trajectory periods.

– Will do (5 days).

Figure 8, right panel: it would be useful to mark the days on the cruise track, to allow a comparison with the left panels.

– We can add these as points.

Figure 9, left panels: the gray symbols are very hard to see on the purple background, please use a different color (e.g. darker). At the end of the caption: "correspond to" instead of "correspond with".

– Will do.

Please clarify early in the paper what "bottom-up" and "top-down" estimates refer to. This terminology in the specific context of the paper may not be clear to every reader. E.g. in Fig. 8, the Authors could add a clarification on "bottom-up" (ship-based) and "top-down" (land-based).

– We can clarify this in the text.

---

## Author Response (AR1)

Dear Dr. Naqvi,

We have made revisions to our manuscript as detailed in our responses to the Reviewers. A track-changes version of our revised manuscript follows, with newly added text in blue and deleted text in red with strikethrough. We have made adjustments to the language where requested, and revised our presentation and discussion of the model uncertainty and limitations, with a particularly focus on the dilution rate constant. We now present two estimates of the flux density, the first being our original approach with a dilution rate constant estimated from in situ data, and the second being flux densities using a dilution rate constant tuned to best match the shipboard flux densities from our research cruise. We have also put our results into broader context through an expanded and revised discussion section. Further changes include: selection of different piston velocity formulations, use of the locally interpreted alkalinity regression (LIAR) for estimation of total alkalinity, and several adjustments to the figures to improve the clarity of data presentation.

Thank you for the opportunity to improve our work.

Kind Regards,
The Authors

[revised manuscript text omitted]

---

## Author Response (AR2)

*Note to the Editor: Please find enclosed a point-by-point reply to the Referee, and a track-changes version of the revised manuscript. Author responses are blue and italicized.*

**Report 1 from Anonymous Referee 2**

This is my second review of the manuscript by Morgan et al. As stated in my first review, this is a nice study with relevance for the Biogeosciences readership. I commend the Authors for the serious consideration that they gave to my and another Reviewer comments. As a result, several aspects of the study have improved, including the presentation of the model, the discussion of uncertainty, and the discussion of the results in a broader context, and the figures.

*–We thank the Referee for their time, and their helpful and constructive comments.*

The fact that the top-down estimates still seem to overestimate fluxes compared to the ship-based estimate even after tuning is still somewhat puzzling, but is well discussed in the paper. (CO2 is better matched compared to O2 and especially N2O, see Fig. 8 and Table 2). The Authors could add a comment on this potential low bias in the abstract as well, since it is an issue that may need to be resolved in future studies (e.g. with comparison to more ship based data).

*–We have added language to the abstract clarifying that the top-down method overestimates the flux, that the agreement varied by species, and that we also present tuned flux estimates.*

It's also somewhat puzzling in Fig. 6 that the land-based concentrations are often higher than the ship-based concentrations (especially for CH4), and more "spiky", since dilution during transport from upwelling regions should have reduced and smoothed them. A discussion of these discrepancies could be added to Section 3.3.

*–There is some language in the caption that gives a very brief explanation of this, but we have added some additional text to Section 3.3. The land-based atmospheric measurements are more variable because of local atmospheric transport, driven by regional topography and the large temperature contrast between the coast and the interior. This causes a diurnal cycle in the species measured at the station. Additionally, there is always some synoptic-scale variability. Hence, not all of the spikes seen in the NDAO record shown in Figure 6 are due to upwelling events.*

Other than that, I feel that the paper is now suitable for publication. As a final suggestion, I encourage the Authors to give a careful reading to the manuscript to catch some lingering typos and awkwardly phrased sentences. A few are listed in the following:

*–The paper has been proofread again, and the errors identified by the Referee have been corrected.*

Page 1, line 16: episodic could be changed to variable, since it refers to both occurrence (which can be episodic), and intensity (which can be variable, rather than episodic). Or alternatively the sentence can be rephrased.

Page 2 , line 24: change based to based on

Page 5, line 10: perhaps use applied instead of generated?

Page 6, line 10: change no flux the gas to no flux of the gas

Page 9, line 15: change 21 of excluded events were filtered based on to Of these, 21 events were excluded based on

Page 11. line 4: should the uncertainty be added to the average flux density of CH4?

Page 12, line 13: change reasonably to reasonably well

Page 12, line 33: change input to parameter

Page 13, line 6: tune the model : please clarify, e.g. use a value of q so that F matches the bottom up value.

Page 15, line 34: change site to sites

Table 1: maybe add a note to clarify the difference between the non-tuned" values and the tuned values

Figure 9, at the end of the caption: correspond to instead of correspond with.

Figure 8, caption: three should now be two?

[revised manuscript text omitted]